# USP39 promotes antiviral defense through post-transcriptional control of RIG-I and stabilization of STING

Jiazheng Quan[1,2☉], Xibao Zhao[1☉], Shaoying Chen[1], Hongrui Li[3], Wei Chen[1], Qianqian Di[1], Xunwei Li[1], Jiajing Zhao[1], Han Wu[1], Jin Chen[1], Yue Xiao[1], Zherui Wu[1], Weilin Chen[1]*

1 Guangdong Provincial Key Laboratory of Infection Immunity and Inflammation, Marshall Laboratory of Biomedical Engineering, Institute of Biological Therapy, Shenzhen University Medical School, Shenzhen University, Shenzhen, China, 2 Institute of Biopharmaceutical and Health Engineering, Tsinghua Shenzhen International Graduate School, Tsinghua University, Shenzhen, China, 3 Institute of Immunology, Zhejiang University School of Medicine, Zhejiang University, Hangzhou, China

☉ These authors contributed equally to this work.
* cwl@szu.edu.cn

## Abstract

RIG-I and STING are critical for mediating the RIG-I and cGAS-STING signaling pathways that guard against viral infection. Here, we report that ubiquitin-specific peptidase 39 (USP39) positively regulates the RIG-I and cGAS-STING pathways to induce antiviral innate immunity in vitro and in vivo. The USP39 deficiency impaired the antiviral immune response of macrophages, leading to low type I IFNs expression, and high RNA and (e.g., VSV, H1N1 PR8) DNA virus (e.g., HSV-1) replication. Moreover, USP39-deficient mice were more sensitive to VSV or HSV-1 infection than control mice. Conversely, USP39 overexpression promoted the antiviral immune response. Mechanistically, we found that USP39 regulates RIG-I protein expression by promoting pre-RIG-I mRNA splicing and maturation. In addition, we also revealed that USP39 interacts with and stabilizes STING protein by deubiquitinating K48-linked polyubiquitin of STING at K288. These data show that USP39 positively regulates RNA and DNA-virus-induced RIG-I and cGAS-STING signaling, respectively, by promoting post-transcriptional control of RIG-I and stabilization of STING. These data provide new insights and potential therapeutic targets to control viral infections.

## Introduction

Viral pathogens exhibit divergent pathogenic profiles contingent on their genomic architecture. RNA viruses, including pandemic coronaviruses (SARS-CoV), lentiviruses (HIV-1), filoviruses (Ebola), and orthomyxoviruses (influenza), leverage error-prone RNA-dependent RNA polymerases devoid of proofreading fidelity to achieve accelerated mutagenesis [1–3]. This replicative plasticity drives rapid antigenic drift, facilitating zoonotic spillover events and compromising therapeutic efficacy through

**Data availability statement:** All relevant data are within the paper and its Supporting information files.

**Funding:** This work was supported by grants from the National Natural Science Foundation of China (https://www.nsfc.gov. cn/) (82371788 to WL.C), the Natural Science Foundation of Guangdong Province (http:// gdstc.gd.gov.cn) (2024A1515013162 to WL.C; 2024A1515012851 to XB.Z; 2023A1515012402 to YX), the Shenzhen Science and Technology Innovation Program (http://stic.sz.gov.cn/) (KQTD20210811090219022 to WL.C), Medical Science and Technology Research Foundation of Guangdong Province (https://wsjkw.gd.gov. cn) (A2023209 to XB.Z) and Program for Youzuzhikeyan of Shenzhen University (https:// www.szu.edu.cn) (SZU2024YZZKY003 to WL.C). The funders had no role in study design, data collection and analysis, decision to publish, or preparation of the manuscript.

**Competing interests:** The authors have declared that no competing interests exist.

**Abbreviations:** BMDMs, bone marrow-derived macrophages; CARDs, caspase activation and recruitment domains; cGAMP, cyclic GMP-AMP; cGAS, cyclic GMP-AMP synthase; CTD, carboxy-terminal domain; ER, endoplasmic reticulum; IFN, interferon; IRF3, IFN-regulatory factor 3; ISGs, IFN-stimulated genes; LGP2, laboratory of genetics and physiology 2; MDA5, melanoma differentiation-associated protein 5; PAMPs, pathogen-associated molecular patterns; PRRs, pattern-recognition receptors; RIG-I, retinoic acid-inducible gene; RLRs, RIG-I-like receptors; siRNA, small interfering RNA; STING, stimulator of interferon genes; TBK1, TANK-binding kinase 1; USPs, ubiquitin-specific peptidases; USP39, ubiquitin-specific peptidase 39; VSV, vesicular stomatitis virus.

drug-resistant variant emergence [4,5]. Meanwhile, hepatitis virus also represents a large danger to public health, especially blood-borne hepatitis, including HBV, is responsible for the high global morbidity and death rate [6]. As such, effective preventive measures and treatments for viral infections must be continually developed and updated.

The innate immune system is the first line of defense against pathogen invasion: pattern-recognition receptors (PRRs) recognize pathogen-associated molecular patterns (PAMPs) and initiate an immune response to recognize and control virus infection [7]. Generally speaking, RNAs and DNAs are recognized by the RIG-I-MAVS and cGAS-STING pathways through retinoic acid-inducible gene (RIG-I)-like receptors (RLRs) and cytoplasmic DNA receptor-Cyclic GMP-AMP synthase (cGAS), respectively. These pathways mediate the induction of type I interferons and activate downstream adaptors that help to combat RNA or DNA viral infection, respectively [8,9].

Focusing first on RNA virus detection and control: The RLR protein family has three members, RIG-I, melanoma differentiation-associated protein 5 (MDA5), and laboratory of genetics and physiology 2 (LGP2). RLRs are primarily localized in the cytosol [10,11], where they detect immunostimulatory RNAs via their central helicase domain and carboxy-terminal domain (CTD). Only RIG-I and MDA5 mediate the transcriptional induction of anti-viral components, as a result of the activity of their amino-terminal caspase activation and recruitment domains (CARDs) [8,12]. In terms of hierarchy, LGP2 regulates RIG-I and MDA5 signaling [13]. Meanwhile, RIG-I recognizes dsRNA or the triphosphate (PPP) of RNA 5′ ends [8], allowing it to respond to poly (I:C) or RNA viruses, such as vesicular stomatitis virus (VSV), Sendai virus, and Newcastle disease virus [14]. Upon detecting and binding RNA, RIG-I interacts with MAVS, an adaptor protein that contains an N-terminal CARD-like structure [13,15–18], activating TANK-binding kinase 1 (TBK1), and then phosphorylates IFN-regulatory factor 3 (IRF3) [8,19,20]. Phosphorylated IRF3 dimerizes and translocates to the nucleus to induce type I interferon (IFN) transcription [21]. Type I IFNs in turn activate IFN-stimulated genes (ISGs) to control viral infection [22].

In terms of detecting and controlling DNA viruses, cGAS is an important sensor for detecting foreign, cytoplasmic DNAs. Upon binding to dsDNA or DNA-RNA hybrids [23], cGAS catalyzes the synthesis of cyclic GMP-AMP (cGAMP), a secondary messenger that activates stimulator of interferon genes (STING) [24,25]. STING is an endoplasmic reticulum (ER)-located protein that binds to cGAMP and promotes itself oligomerization so that it can recruit and interact with iRhom2 and the Sec5/TRAPβ/Sec61β translocon complex, and then translocate to ER-Golgi intermediate compartments [26,27]. Then STING recruits TBK1 and activates the phosphorylation of IRF3, which dimerizes and translocates to the nucleus to induce the transcription of type I IFNs [28].

Ubiquitin-specific peptidases (USPs) family belong to deubiquitinases with more than 50 members in humans and play wide and profound effect on the regulation of multiple biological processes [29]. Interestingly, in this work, we found the type I IFNs were down-regulated in ubiquitin-specific peptidase 39 (USP39) deficient

macrophages of mice after infected with RNA or DNA virus. USP39 splices pre-mRNAs and deubiquitinates proteins. It was originally thought that USP39 only performed splicing functions, because of the lack of conserved sites of cysteine, histidine, and aspartic acid in the deubiquitinating domain [30]. Our previous study also found that USP39 stabilizes IκBα to regulate NF-κB pathway-mediated inflammatory responses via its deubiquitinating function [31]. However, USP39 is little known on viral infection and antiviral innate immune response, especially the role and mechanism remain unclear, and thus hinders progress in USP39 involves the immune response to virus infection.

Here, we aimed to uncover how the USP39 regulates the antiviral immune response in RNA and DNA viral infection. Our results demonstrated that USP39 enhances anti-RNA and DNA viral immune responses by promoting RIG-I mRNA maturation and STING protein stabilization via splicing pre-mRNAs and deubiquitinating functions, respectively. These results provide new insights and highlight USP39s regulation of antiviral immune responses.

## Results

### USP39 promotes innate immune response against RNA viral infection

We first aimed to investigate whether USP39 has an anti-RNA viral role in macrophages against RNA viral infection. To address this question, we infected *Usp39*-deficient (*Usp39^{fl/fl} Lyz2 Cre*) and control (*Usp39^{fl/fl} Lyz2*) macrophages with either VSV or H1N1 PR8 and then observed replication of virus and antiviral immune responses in cells. *VSV-N* and *H1N1 PR8* mRNA levels were significantly increased in *Usp39^{fl/fl} Lyz2 Cre* macrophages (Fig 1A and 1B). We then infected these macrophages with VSV-eGFP and analyzed expression levels by flow cytometry. The percentage of GFP-positive macrophages was higher significantly in *Usp39^{fl/fl} Lyz2 Cre* macrophages than *Usp39^{fl/fl}* macrophages (Fig 1C). We then collected the supernatants of macrophages infected with VSV and analyzed them by TCID50 assay. The TCID50 counts of *Usp39^{fl/fl} Lyz2 Cre* macrophages were significantly higher than *Usp39^{fl/fl}* macrophages (Fig 1D), indicating that RNA viral replication was increased in *Usp39*-deficient macrophages.

To further investigate the role of USP39 in mediating RNA viral infection, we overexpressed an USP39 plasmid in HeLa cells (S1A Fig) and then infected the cells again with VSV or H1N1 PR8. *VSV-N* and *H1N1 PR8* mRNA levels were significantly decreased in HA-USP39 HeLa cells compared to control (empty vector) cells (Fig 1E and 1F). We confirmed these findings by fluorescence microscopy based on VSV-eGFP expression (S1B Fig). These data suggest that USP39 inhibits RNA virus replication in vitro.

Next, we investigated if USP39 expression affected the RNA virus-triggered type I IFNs response. Indeed, we found that the *Usp39^{fl/fl} Lyz2 Cre* macrophages infected with VSV expressed significantly less *Ifn-β* and *Ifn-α4* mRNA than control macrophages (Fig 1G). IFN-β protein levels (detected from cell supernatants by ELISA) were also decreased in *Usp39^{fl/fl} Lyz2 Cre* versus control macrophages (Fig 1H). We obtained similar results for *Ifn-β* upon H1N1 PR8 infection (Fig 1I).

We also transfected macrophages with Poly (I:C)—a synthetic dsRNA analog that can activate RIG-I signaling [32]. USP39-deficient macrophages also expressed lower *Ifn-β* and *Ifn-α4* mRNA and IFN-β protein levels compared to control macrophages (Fig 1J and 1K).

To ensure that type I IFNs-stimulated genes were induced successfully, we detected the mRNA levels of *Isg15*, *Isg56*, and *Cxcl10* - which are type I IFNs targets—by qPCR. The expression of these mRNAs was also decreased in *Usp39^{fl/fl} Lyz2 Cre* macrophages compared with *Usp39^{fl/fl}* macrophages after VSV infection and Poly (I:C) transfection (Fig 1L and 1M). Meanwhile, *Ifn-β* mRNA levels increased in HA-USP39 overexpressing HeLa cells infected with VSV or H1N1 PR8 compared to the negative control, VSV or H1N1 PR8 infected cells (Figs 1N and S1C).

We previously showed that USP39 stabilizes IκBα to regulate the NF-κB signaling pathway [31]. Similarly, USP39 deficiency induced inflammatory overexpression in viral infection. Indeed, we found that *Il-6* mRNA—which is an inflammatory marker and activated by NF-κB signaling pathway—was raised in *Usp39^{fl/fl} Lyz2 Cre* macrophages after VSV infection

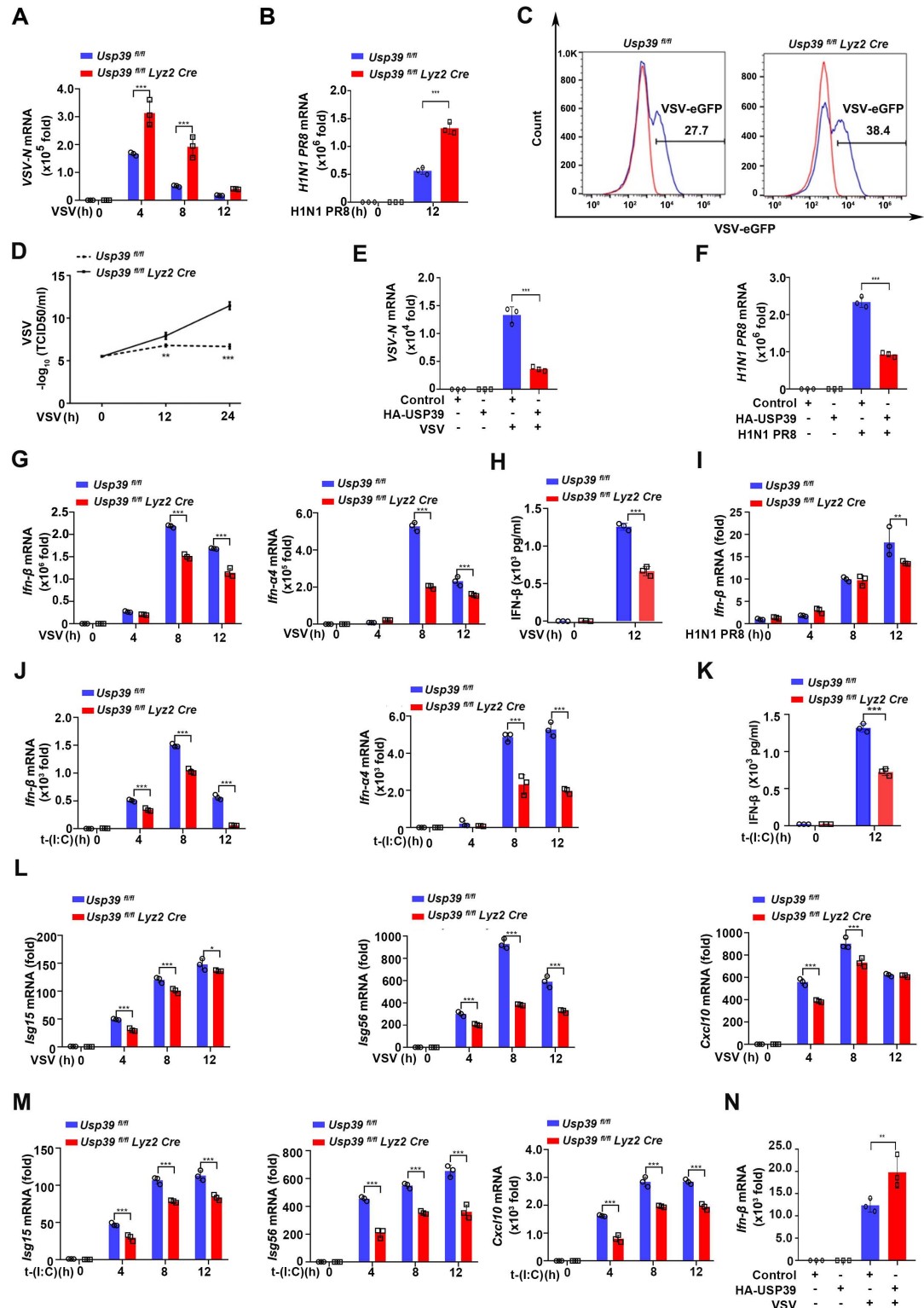

**Fig 1. USP39 promotes innate immune response against RNA viral infection. (A, B)** *Usp39*fl/fl and *Usp39*fl/fl *Lyz2 Cre* macrophages were infected with VSV (MOI = 1) (A) or H1N1 PR8 (MOI = 1) (B) for the indicated time, and viral replication was measured by qPCR. **(C)** *Usp39*fl/fl and *Usp39*fl/fl *Lyz2*

*Cre* macrophages were infected with VSV-eGFP (MOI = 1), and the percentage of GFP⁺ macrophages was detected by flow cytometry. **(D)** *Usp39*fl/fl and *Usp39*fl/fl *Lyz2 Cre* macrophages were infected with VSV (MOI = 1) for 12 or 24 h, and VSV loads were measured by TCID50 assay. **(E, F)** HeLa cells overexpressing a Control vector or HA-USP39 were infected with VSV (MOI = 1) (E) or H1N1 PR8 (MOI = 1) (F) for 12 h before viral replication was measured by qPCR. **(G, H)** *Usp39*fl/fl and *Usp39*fl/fl *Lyz2 Cre* macrophages were infected with VSV (MOI = 1) for the indicated time. *Ifn-β* and *Ifn-α4* mRNA were measured by qPCR (G), and supernatants were collected and IFN-β proteins were measured by ELISA (H). **(I)** *Usp39*fl/fl and *Usp39*fl/fl *Lyz2 Cre* macrophages were infected with H1N1 PR8 (MOI = 1) for the indicated time. *Ifn-β* mRNA was measured by qPCR. **(J, K)** *Usp39*fl/fl and *Usp39*fl/fl *Lyz2 Cre* macrophages were transfected with Poly (I:C) (1 µg/mL) for the indicated time. *Ifn-β* and *Ifn-α4* mRNA were measured by qPCR (J), and supernatants were collected and IFN-β proteins were measured by ELISA (K). **(L, M)** *Usp39*fl/fl and *Usp39*fl/fl *Lyz2 Cre* macrophages were infected with VSV (MOI = 1) (L) or transfected with Poly (I:C) (1 µg/mL) (M) for the indicated time. Isg*15*, *Isg56*, and *Cxcl10* mRNA were measured by qPCR. **(N)** HeLa cells overexpressing a control vector or HA-USP39 were infected with VSV (MOI = 1) for 12 h. *Ifn-β* mRNA was measured by qPCR. The data represent the means ± SD, from three independent experiments. *$p < 0.05$, **$p < 0.01$, ***$p < 0.001$ using Student *t* test. This da*ta* underlying this Figure can be found in S1 Data and S1 Raw Images.

(S1D Fig). Taken together, these results showed that USP39 positively regulates the type I IFNs response as a cellular defense against RNA viral infection.

## USP39 promotes innate immune response against DNA viral infection

We next evaluated the impact of USP39 on the macrophage response against DNA viral infection. For these experiments, we infected *Usp39*^fl/fl *Lyz2 Cre* macrophages with HSV-1 and HSV-2. *Usp39*^fl/fl *Lyz2 Cre* macrophages showed higher *HSV-1 UL30* and *HSV-2 ICP27* expression compared with *Usp39*^fl/fl macrophages after HSV-1 or HSV-2 infection (Fig 2A and 2B). Moreover, *HSV-1 UL30* or *HSV-2 ICP27* mRNA expression decreased in infected, HA-USP39 overexpressing HeLa cells compared with control cells (Fig 2C and 2D). We thus posit that USP39 inhibits DNA viral replication.

Next, we saw that *Ifn-β* and *Ifn-α4* mRNA and IFN-β protein levels were significantly decreased in *Usp39*^fl/fl *Lyz2 Cre* macrophages infected with HSV-1 (Fig 2E and 2F). Double-stranded DNA (dsDNA), a DNA virus mimic, can activate the cGAS-STING signaling pathway [33]. Intracellular transfection of dsDNA also resulted in reduced *Ifn-β* and *Ifn-α4* mRNA and IFN-β protein levels in *Usp39*^fl/fl *Lyz2 Cre* macrophages (Fig 2G and 2H). *Isg15*, *Isg56*, and *Cxcl10* mRNA levels were also decreased in *Usp39*^fl/fl *Lyz2 Cre* macrophages compared with *Usp39*^fl/fl macrophages after HSV-1 infection or dsDNA transfection (Fig 2I and 2J). Finally, *Il-6* mRNA levels increased in *Usp39*^fl/fl *Lyz2 Cre* macrophages after HSV-1 infection (S1E Fig). These data support that USP39 positively regulates the type I IFNs response as a cellular defense against DNA viral infection.

## *Usp39* deficiency impairs antiviral immune responses in vivo

To investigate the role of USP39 in the host defense against viral infection in vivo, we infected *Usp39*^fl/fl *Lyz2 Cre* and *Usp39*^fl/fl mice with VSV or HSV-1 via tail vein or intraperitoneal injection. *Ifn-β* and *Ifn-α4* mRNA levels in the spleen, liver and lung and IFN-β protein levels in the sera were significantly decreased in *Usp39*^fl/fl *Lyz2 Cre* mice, 12 h after VSV infection (Fig 3A–3D). In addition, we saw higher levels of VSV replication detected by TCID50 in the spleen, liver and lung (Fig 3E–3G), and correspondingly lower survival within four days (Fig 3H) in *Usp39*^fl/fl *Lyz2 Cre* mice infected with VSV compared with littermate controls. Moreover, H&E staining of the lungs indicated more severe inflammation in VSV-infected *Usp39*^fl/fl *Lyz2 Cre* mice compared with control mice (Fig 3I). We obtained similar data when infecting the mice with HSV-1, *Ifn-β* and *Ifn-α4* mRNA levels in the spleen, liver, and lung and IFN-β protein levels in the sera were significantly decreased in *Usp39*^fl/fl *Lyz2 Cre* mice, 12 h after HSV-1 infection (Fig 3J–3M). Furthermore, TCID50 assays showed elevated HSV-1 replication levels in the spleen, liver, and lung (Fig 3N–3P). Correspondingly, HSV-1-infected *Usp39*^fl/fl *Lyz2 Cre* mice exhibited significantly lower survival rates within 20 hours compared with their littermate controls (Fig 3Q), and H&E staining of the lungs further revealed more severe inflammation in these mice relative to controls (Fig 3R). Taken together, these data indicate that *Usp39*^fl/fl *Lyz2 Cre* mice are more susceptible to VSV and HSV-1 infection in vivo than *Usp39*^fl/fl, littermate control mice.

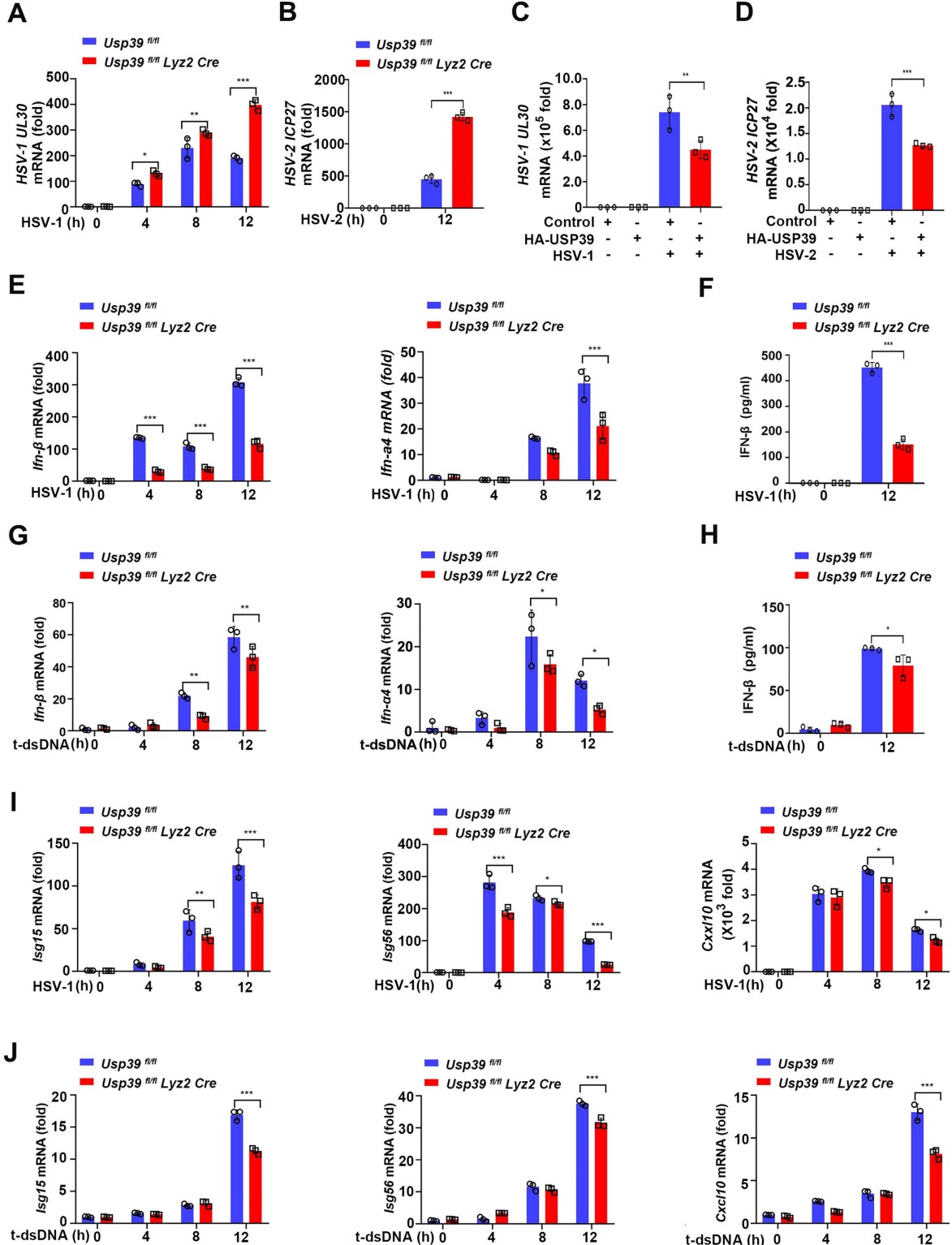

**Fig 2. USP39 promotes innate immune response against DNA viral infection. (A, B)** *Usp39*fl/fl and *Usp39*fl/fl *Lyz2 Cre* macrophages were infected with HSV-1 (MOI = 10) (A) or HSV-2 (MOI = 10) (B) for the indicated time, and viral replication was measured by qPCR. **(C, D)** HeLa cells overexpressing

a control vector or HA-USP39 were then infected with HSV-1 (MOI = 10) (C) or HSV-2 (MOI = 10) (D) for 12 h before viral replication was measured by qPCR. **(E, F)** *Usp39*fl/fl and *Usp39*fl/fl *Lyz2 Cre* macrophages were infected with HSV-1 (MOI = 10) for the indicated time. *Ifn-β* and *Ifn-α4* mRNA were measured by qPCR (E), and supernatants were collected and IFN-β proteins were measured by ELISA (F). **(G, H)** *Usp39*fl/fl and *Usp39*fl/fl *Lyz2 Cre* macrophages were transfected with dsDNA (10 μg/mL) for the indicated time. *Ifn-β* and *Ifn-α4* mRNA were measured by qPCR (G), and supernatants were collected and IFN-β proteins were measured by ELISA (H). **(I, J)** *Usp39*fl/fl and *Usp39*fl/fl *Lyz2 Cre* macrophages were infected with HSV-1 (MOI = 10) (I) or transfected with dsDNA (10 μg/mL) (J) for the indicated time, *Isg15*, *Isg56*, and *Cxcl10* mRNA were measured by qPCR. The data represent the means ± SD, from three independent experiments. *$p < 0.05$, **$p < 0.01$, ***$p < 0.001$ using Student *t* test. This da*ta* underlying this Figure can be found in S1 Data and S1 Raw Images.

## USP39 regulates antiviral immune response by targeting RIG-I and STING, respectively

To explore the mechanism by which USP39 regulates the antiviral response to RNA and DNA viruses, we analyzed components of the RIG-I-MAVS and cGAS-STING signaling pathways by western blotting. First, we found that basal RIG-I protein levels were downregulated, causing weaker RIG-I stimulation from 0 to 8 h after *Usp39*<sup>fl/fl</sup> *Lyz2 Cre* macrophages were infected with VSV or transfected with Poly (I:C) (Fig 4A and 4B). MAVS (Mitochondrial Antiviral Signaling Protein) is a core adaptor protein in the RIG-I-like receptor signaling pathway during RNA virus-induced innate immune responses. It integrates upstream viral RNA recognition signals via its mitochondrial localization, transmits the signals downstream to activate antiviral immune responses, and serves as a key molecular hub in the innate immune defense against RNA viruses. Our western blotting result also revealed that overexpression of different doses of USP39 did not influence the protein level of MAVS in 293T cells (S1H Fig). Furthermore, to further confirm this result, we also detected the MAVS mRNA expression, and the result also demonstrated that USP39 did not regulate the mRNA expression of MAVS (S1I Fig). We then transiently overexpressed HA-USP39 in HEK293T cells. Accordingly, endogenously express RIG-I increased in these cells compared to controls (Fig 4C). We thus hypothesize that USP39 targets the RIG-I protein.

We next analyzed cGAS-STING signaling pathway, and noted a decrease of STING protein in *Usp39*<sup>fl/fl</sup> *Lyz2 Cre* macrophages infected with HSV-1 or transfected with dsDNA compared with control macrophages (Fig 4D and 4E). Because STING activation is dependent on cGAMP stimulation, we transfected macrophages with cGAMP (1 μg/mL) at the indicated times to see if USP39 regulates the STING-mediated signaling pathway. Here, we saw that *Usp39*<sup>fl/fl</sup> *Lyz2 Cre* macrophages expressed lower STING protein and in turn, exhibited weaker p-TBK1 stimulation than control macrophages (Fig 4F). *Usp39*<sup>fl/fl</sup> *Lyz2 Cre* macrophages also expressed lower *Ifn-β*, *Ifn-α4* (Fig 4G), *Isg15*, *Isg56*, and *Cxcl10* mRNA (S1F Fig) compared with control macrophages. Finally, upon transiently transfecting HA-USP39 into HEK293T cells, we saw that HA-USP39 overexpression increased endogenously STING protein levels (Fig 4H). Moreover, we overexpressed Flag-RIG-I, Flag-STING, Flag-TBK1 in HEK293T cells, respectively. The Flag-TBK1 as a negative control, which indicated that Flag-RIG-I and Flag-STING were affected by USP39 overexpression (Fig 4I). We further performed ubiquitination assays to detect the effect of Myc-USP39 on regulating Flag-RIG-I protein ubiquitination (MG132 treated, equal RIG-I protein amount). The results showed that USP39 could not regulate the ubiquitination level of RIG-I (Fig 4J), indicating that USP39 regulates RIG-I not at the post-translational protein level. Taken together, the data indicate that USP39 positively regulates antiviral immune responses by targeting RIG-I and STING.

## USP39 promotes RIG-I mRNA maturation

USP39 is involved in protein deubiquitination proteins and pre-mRNA splicing [31,34–39]. We therefore investigated whether USP39 affects *Rig-i* mRNA maturation. Our data showed that *Rig-i* mRNA levels were decreased in *Usp39* knockdown macrophages (Figs 5A and S1G) and *Usp39*<sup>fl/fl</sup> *Lyz2 Cre* macrophages (Fig 5B), but increased in HA-USP39 overexpressing HeLa cells (Fig 5C). Furthermore, and consistent with our protein findings (Fig 4A and 4B), *Rig-i* mRNA was also downregulated in *USP39-deficient* macrophages after VSV infection or poly (I:C) transfection (Fig 5D and 5E).

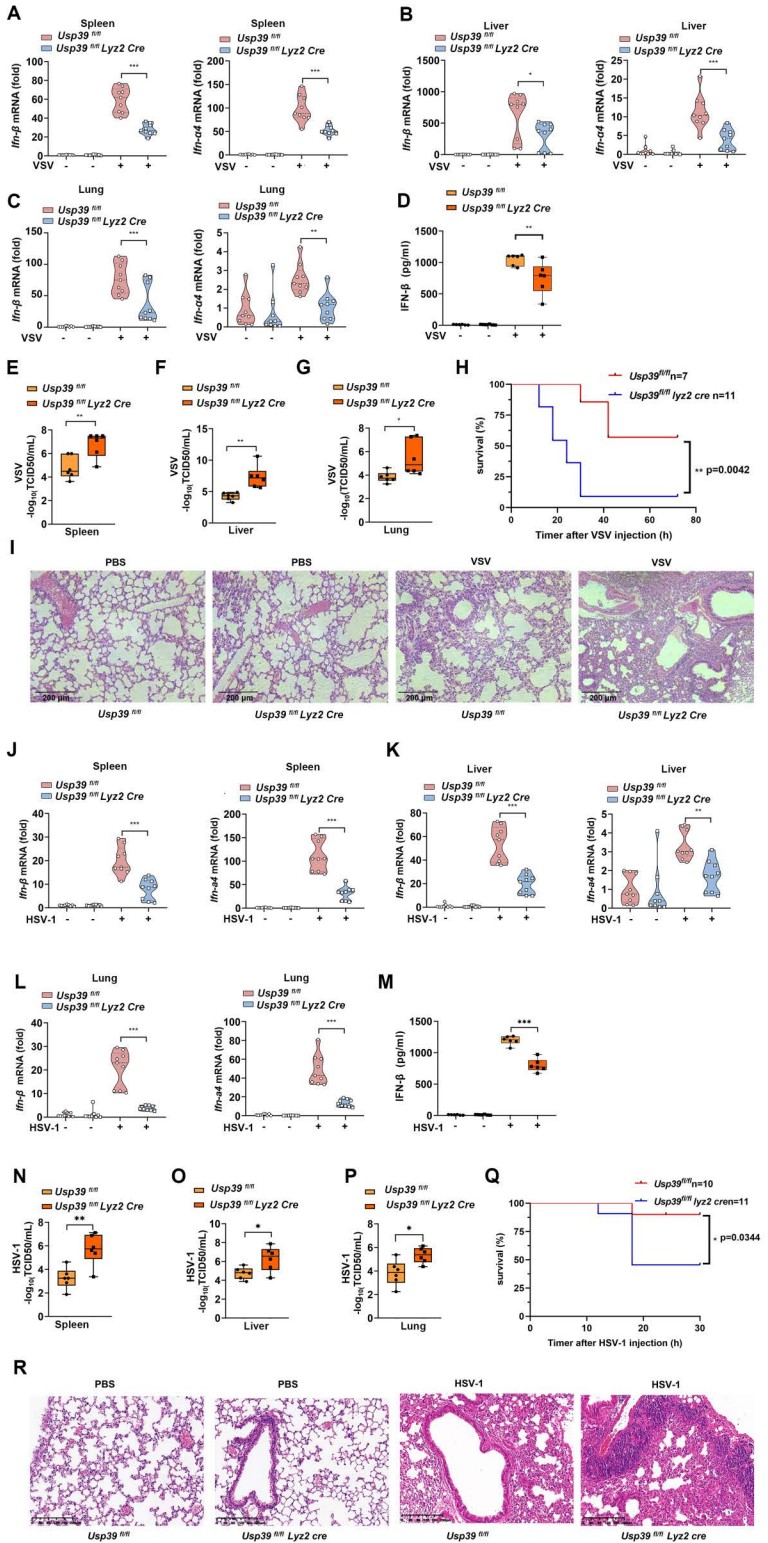

**Fig 3. *Usp39* deficiency impairs antiviral immune responses in vivo. (A–G)** *Usp39*fl/fl and *Usp39*fl/fl *Lyz2 Cre* mice (*n* = 9 per group) were administered VSV via the tail vein (250 μl, 1 × 10⁹/g) for 12 h. *Ifn-β* and *Ifn-α4* mRNA levels in the spleen (A), liver (B) and lung (C) were measured by qPCR and IFN-β protein in sera was measured by ELISA (D). VSV loads in the spleen (E), liver (F) and lung (G) were detected by TCID50 assay. **(H)** Survival of

*Usp39*fl/fl and *Usp39*fl/fl *Lyz2 Cre* mice exposed to VSV via tail vein injection (350μl, 1x10⁹/g) (*n* = 7−11 per group). Statistical significance was calculated using the Long-rank (Mantel-Cox) test. **(I)** H&E analysis of *Usp39*fl/fl and *Usp39*fl/fl *Lyz2 Cre* mice exposed to VSV. **(J–P)** *Usp39*fl/fl and *Usp39*fl/fl *Lyz2 Cre* mice (*n* = 9 per group) were exposed to HSV-1 via tail vein (250 μl, 1 × 10⁸/g) or intraperitoneal injection (1 ml, 1 × 10⁸/g) for 12 h. *Ifn-β* and *Ifn-α4* mRNA levels in the spleen (J), liver (K), and lung (L) were measured by qPCR and IFN-β protein in sera was measured by ELISA (M). HSV-1 loads in the spleen (N), liver (O), and lung (P) were detected by TCID50 assay. **(Q)** Survival of *Usp39*fl/fl and *Usp39*fl/fl *Lyz2 Cre* mice exposed to HSV-1 via intraperitoneal injection (1.5 ml, 1 × 10⁸/g) (*n* = 10−11 per group). Statistical significance was calculated using the Long-rank (Mantel-Cox) test. **(R)** H&E analysis of *Usp39*fl/fl and *Usp39*fl/fl *Lyz2 Cre* mice exposed to HSV-1. The data represent the means ± SD. *$p < 0.05$, **$p < 0.01$, ***$p < 0.001$ using Student *t* test. This da*ta* underlying this Figure can be found in S1 Data and S1 Raw Images.

We used the sequence of the region from exon 3–4 in the human *Rig-i* primary transcript and exon 2–3 in the mouse *Rig-i* primary transcript as specific primers to detect spliced (exon-exon) and unspliced (exon-intron) *Rig-i* mRNA (S2A and S2B Fig). The results of our qPCR analysis showed that unspliced *Rig-i* mRNA transcripts were increased and spliced *Rig-i* mRNA transcripts were decreased in *Usp39*^*fl/fl*^ *Lyz2 Cre* macrophages compared with *Usp39*^*fl/fl*^ macrophages (Fig 5F). Meanwhile, unspliced *Rig-i* mRNA transcripts were decreased in HeLa cells overexpressing HA-USP39 compared with control HeLa cells, whereas spliced *Rig-i* mRNA levels were increased (Fig 5G). We obtained consistent results when comparing *Usp39*^*fl/fl*^ *Lyz2 Cre* macrophages with control macrophages that were transfected with Poly (I:C) (S2C Fig) and HeLa cells that overexpressed HA-USP39 and were then infected with VSV (S2D Fig). Given that mouse RIG-I contains 18 exons, we examined the ratio of its exons to the 18 exons-introns junctions in bone marrow-derived macrophages (BMDMs). Our results demonstrated that this ratio was significantly decreased in USP39-deficient BMDMs, indicating that USP39 is involved in *Rig-i* gene splicing (S2E Fig).

To confirm that USP39 regulates RIG-I expression by affecting its mRNA maturation, we co-overexpressed HA-USP39 in HEK293T cells and then performed an RNA-binding protein immunoprecipitation assay (RIP). PCR and qPCR analysis showed that HA-USP39 binds the pre-mRNA of RIG-I to allow it to mature, and the *RIG-I* pre-mRNA was significantly enriched in USP39 pull-down assay (Fig 5H).

To further understand USP39 deficiency impairs the anti-RNA viral responding of macrophages, we stably overexpressed *Usp39* or a control vector in *Usp39*^*fl/fl*^ *Lyz2 Cre* BMDMs by lentiviral transduction [31]. *Ifn-β* mRNA expression was recovered, and VSV replication decreased in infected *Usp39*^*fl/fl*^ *Lyz2 Cre* USP39-overexpressing BMDMs compared to infected control BMDMs (Fig 5I and 5J). Furthermore, when USP39 was restored, the decreased levels of *Rig-i* I mRNA and the exon-to-intron ratio were also recovered (Fig 5K). To check if the deubiquitinating function of USP39 helps to regulate RIG-I, we transfected Myc-tagged wild-type USP39 and a Myc-tagged USP39-CA enzyme inactive mutant (USP39-C306A) into HeLa cells and then infected the cells with VSV. Interestingly, *Rig-i* mRNA still increased in Myc-USP39-CA overexpressing cells compared with Myc-USP39 overexpressing cells, and the splicing efficiency of *Rig-i* was not affected by USP39-CA (Fig 5L). To further explore the underlying mechanism, we co-overexpressed HA-USP39 and Myc-RIG-I plasmids in cells, and then performed RNA-binding protein immunoprecipitation (RIP) assays. The results demonstrated that the USP39 could bind the mRNA of RIG-I (S2H Fig), suggesting that USP39 might also bind the mature RIG-I mRNA and stabilize it, then promoting the increase of RIG-I. The previous study [40] suggests that USP39 can deubiquitinate and stabilize STAT1. To exclude the effects of USP39 on IFN-IFNAR-JAK/STAT signaling. We get the STAT1-knockout cells, and the results showed that USP39 promoted VSV-induced *Ifn-β* expression and *Rig-i* pre-mRNA splicing (at baseline) in a STAT1-independent manner (Figs 5M, 5N, S2F, and S2G). These results demonstrate that USP39 promotes *Rig-i* pre-mRNA splicing and maturation.

## USP39-mediated K48-linked deubiquitination stabilizes STING

To better understand the relationship between USP39 and STING, we monitored STING mRNA levels in macrophages. Differing from the results were obtained for RIG-I, STING mRNA levels were unchanged in *Usp39*^*fl/fl*^ *Lyz2 Cre* macrophages compared with *Usp39*^*fl/fl*^ macrophages (Fig 6A). We then overexpressed Flag- or His-STING and Myc-USP39 in

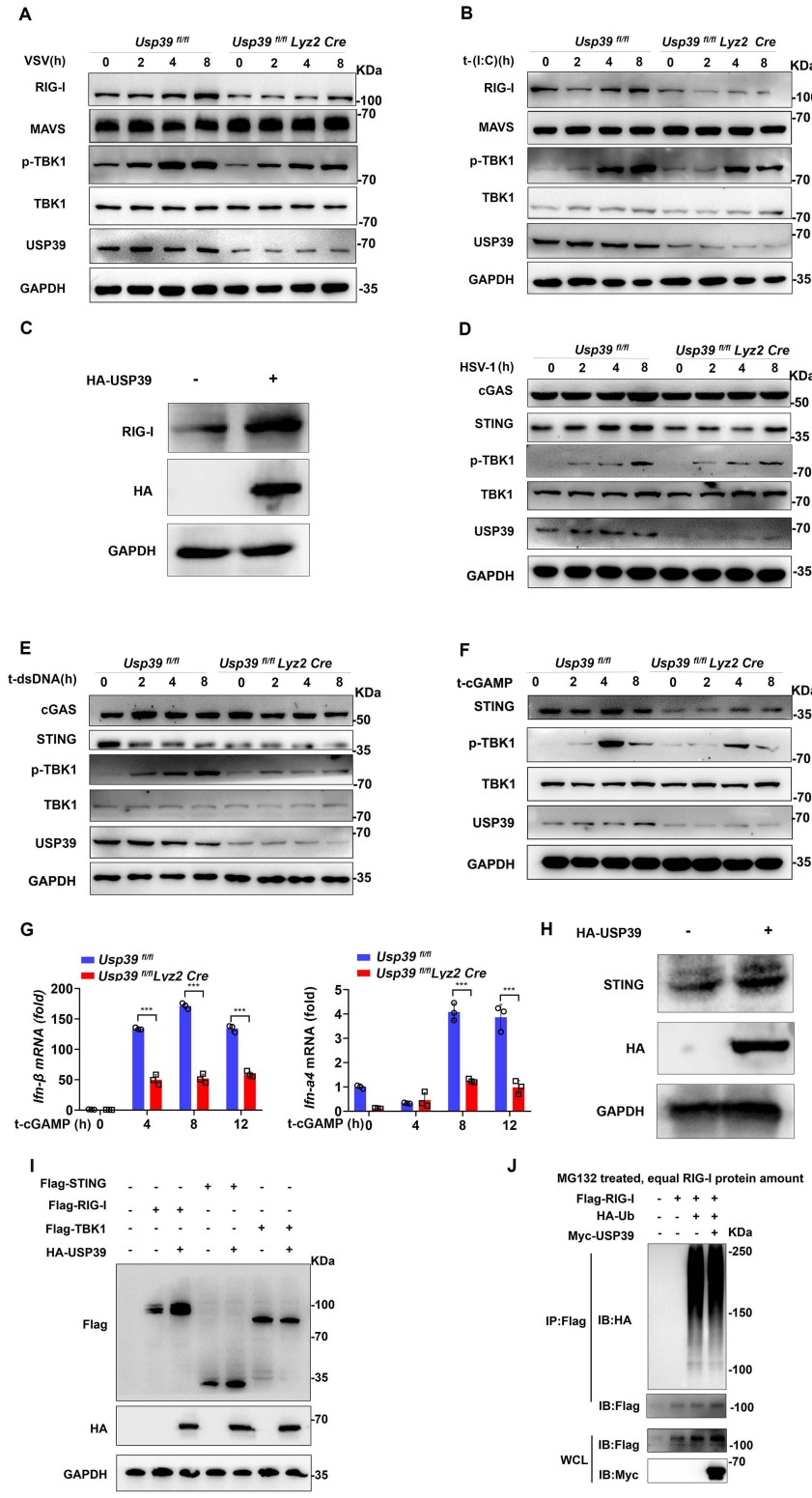

**Fig 4. USP39 regulates antiviral immune response by targeting RIG-I and STING, respectively. (A, B)** western blot analyses of *Usp39*fl/fl and *Usp39*fl/fl *Lyz2 Cre* macrophages infected with VSV (MOI = 1) (A) or transfected with Poly (I:C) (1 μg/mL) (B) for the indicated time. **(C)** HA-USP39 was

overexpressed in HEK293T cells, and cell lysates were analyzed by western blotting. **(D, E)** western blot analyses of *Usp39*fl/fl and *Usp39*fl/fl *Lyz2 Cre* macrophages infected with HSV-1 (MOI = 10) (D) or transfected with dsDNA (10 μg/mL) (E) for the indicated time. **(F, G)** *Usp39*fl/fl and *Usp39*fl/fl *Lyz2 Cre* macrophages were transfected with cGAMP (1 μg/mL) for the indicated time, and cell lysates were immunoblotted with indicated antibodies (F), and *Ifn-β* and *Ifn-α4* mRNA levels were measured by qPCR (G). **(H)** HA-USP39 was overexpressed in HEK293T cells, and cell lysates were analyzed by western blotting. **(I)** Flag-RIG-I, Flag-STING, Flag-TBK1, and HA-USP39 were co-overexpressed in HEK293T cells, respectively. Cell lysates were analyzed by western blotting. **(J)** Flag-RIG-I, HA-Ub, Myc-USP39, were co-overexpressed in HEK293T cells before the cells were treated with MG132 (20 μM) for 6 h. The cell lysates were precipitated with M2 beads and equal RIG-I protein was determined by western blotting. The data represent the means ± SD, from three independent experiments. *$p < 0.05$, **$p < 0.01$, ***$p < 0.001$ using Student *t* test. This data underlying this Figure can be found in S1 Data and S1 Raw Images.

HEK293T cells. Immunofluorescence staining showed that USP39 co-localized with STING (Fig 6B), and co-immunoprecipitation analysis revealed an interaction between the overexpressed proteins (Fig 6C and 6D). Meanwhile, STING interacted more strongly to USP39 depend on higher expression of Myc-USP39 (Fig 6D). We confirmed that the interaction was direct by GST pulldown assay (Fig 6E). We thus infer that USP39 modulates STING function via a direct protein interaction.

Next, results of a cycloheximide chase assay indicated that STING protein degradation was inhibited by overexpression of USP39 in HEK293T cells (Fig 6F). Next, we saw the STING could be rescued by MG132 treatment in *Usp39*^fl/fl^ *Lyz2 Cre* macrophages (Fig 6G). It is likely, therefore, that USP39 stabilizes STING via its deubiquitinating activity. To test this hypothesis, we transiently co-transfected Myc-USP39, Flag-STING, and HA-Ub-WT/K48 in HEK293T cells. Subsequent immunoprecipitation analysis revealed that USP39 over-expression resulted in the removal of K48-linked polyubiquitin from STING (Fig 6H). Conversely, K48-linked STING polyubiquitination was elevated in *Usp39*^fl/fl^ *Lyz2 Cre* macrophages compared with control macrophages (Fig 6I). Given that OTUB1 is highly specific for K48-linked ubiquitin chains and can remove most K48-linked ubiquitin chains on proteins [41,42], we got the Myc-OTUB1 expression plasmid to verify whether the ubiquitin signals detected are covalently conjugated to STING rather than non-covalently associated ubiquitin chains. We performed co-expression assays in 293T cells by transfecting HA-Ub, Flag-STING, and Myc-OTUB1, with or without additional Myc-USP39. Consistent with expectation, immunoprecipitation of HA-Ub followed by immunoblotting for Flag-STING showed that USP39 was markedly reduced ubiquitin chains conjugated STING (S2I Fig). Collectively, these data indicate that USP39 interacts with STING and inhibits its degradation by removing K48-linked polyubiquitin.

**USP39-mediated K48-linked deubiquitination of STING at Lys288 depends on its deubiquitinating enzyme activity**

In our final analyses, we assessed whether the stabilization of STING is dependent on the deubiquitinating enzyme activity of USP39 and monitored the specific sites on the STING protein at which USP39 targets. We first saw that *Ifn-β* mRNA levels recovered and HSV-1 replication decreased in infected *Usp39*^fl/fl^ *Lyz2 Cre* BMDMs overexpressing USP39 (to restore USP39 levels) compared to un-transfected cells (i.e., with no USP39 expression) infected with HSV-1 (Fig 7A and 7B). To confirm the authenticity of USP39 deubiquitinating's function for STING, we co-transfected Myc-USP39 and Myc-USP39-CA in HEK293T cells: Myc-USP39 but not Myc-USP39-CA stabilized the STING protein (Fig 7C) and removed K48-linked polyubiquitin from STING (Fig 7D).

In order to determine the deubiquitinating site of STING targeted by USP39, we then searched the UniProt database for potential lysine sites of STING that might be deubiquitinated by USP39 (https://www.uniprot.org/uniprotkb?query=STING), and found that lysines 235, 288, 369, 377 were conserved across different species (S3A Fig). We mutated these four sites to arginine, and found that only the K288R mutation abolished USP39-mediated deubiquitination of STING (Fig 7E). It's suggesting that USP39 deubiquitinates and stables STING depending on K288 lysine site of STING. Taken together, these data indicate that USP39 deubiquitinating enzyme activity is required to remove K48-linked polyubiquitin from STING at K288 and that this process ensures the activation of a STING-mediated immune response against DNA viral infection.

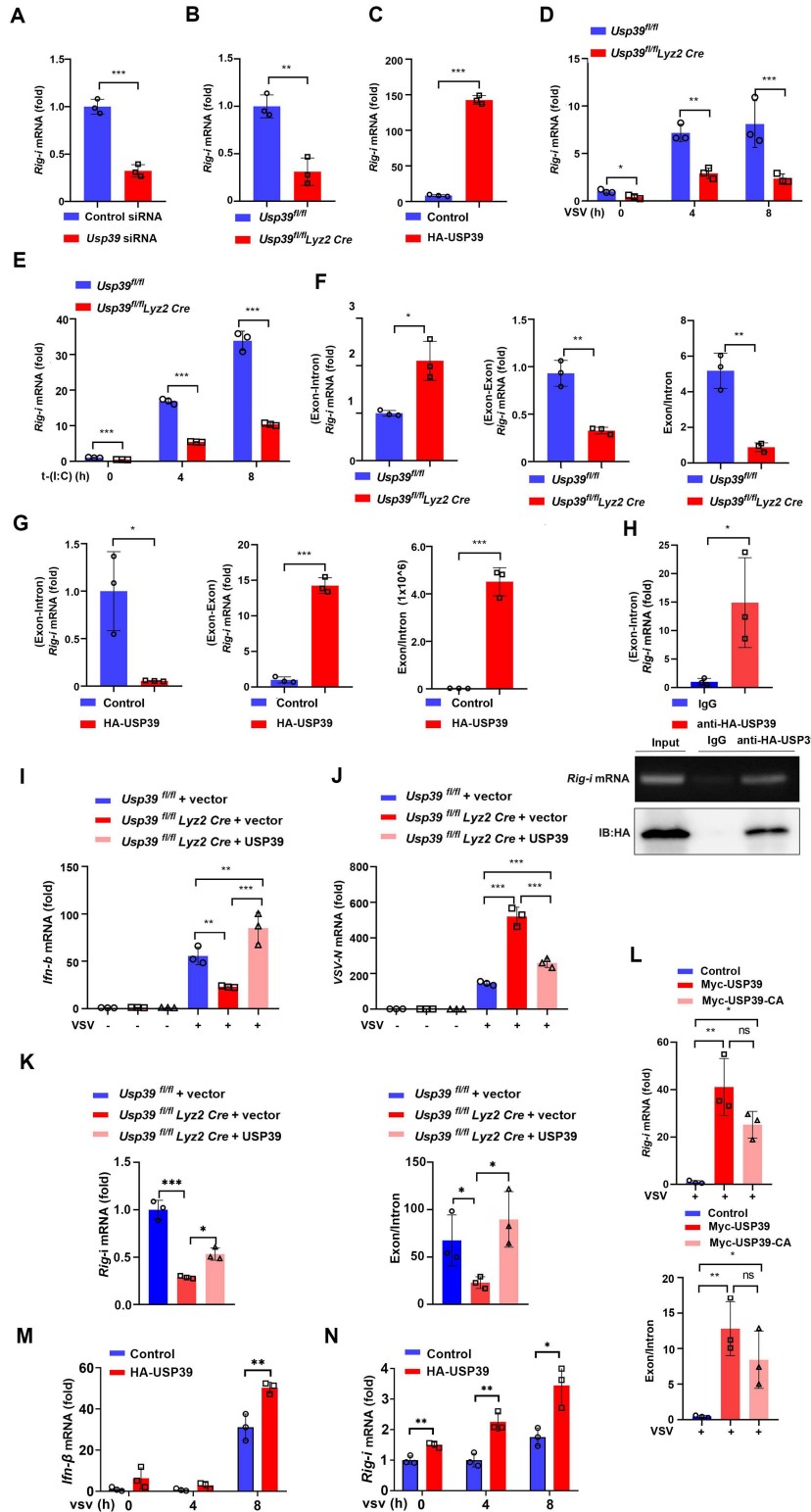

**Fig 5. USP39 promotes RIG-I mRNA maturation. (A–C)** USP39 was knocked down in macrophages using *Usp39* siRNA, and *Rig-i* mRNA levels were measured by qPCR (A). *Rig-i* mRNA levels were measured by qPCR in *Usp39*fl/fl and *Usp39*fl/fl *Lyz2 Cre* macrophages (B). A control vector or

HA-USP39 were overexpressed in HeLa cells, and *RIG-I* mRNA levels were measured by qPCR (C). **(D, E)** *Usp39*fl/fl and *Usp39*fl/fl *Lyz2 Cre* macrophages were infected with VSV (MOI = 1) (D) or transfected with Poly (I:C) (1 μg/mL) (E) for the indicated time. *Rig-i* mRNA levels were measured by qPCR. **(F)** Special *Rig-i* mRNA levels in *Usp39*fl/fl and *Usp39*fl/fl *Lyz2 Cre* macrophages were measured by qPCR. **(G)** A control vector or HA-USP39 were overexpressed in HeLa cells before Special *Rig-i* mRNA levels were measured by qPCR. **(H)** HA-USP39 was overexpressed in HEK293T cells and performed by RIP method, *Rig-i* mRNA levels measured by PCR (lower) or qPCR (upper), and HA tag was analyzed by western blotting. **(I–K)** USP39 was overexpressed in *Usp39*fl/fl and *Usp39*fl/fl *Lyz2 Cre* BMDMs by lentiviral transduction before the cells were infected with VSV (MOI = 1) for 12 h. *Ifn-β* (I), *VSV-N* (J) mRNA levels, *Rig-i* mRNA, and splicing efficiency (K) were measured by qPCR. **(L)** A control vector, Myc-USP39 or Myc-USP39-CA were overexpressed in HeLa cells before the cells were infected with VSV for 12 h. *RIG-I* mRNA and splicing efficiency were then measured by qPCR. **(M, N)** HA-USP39 was overexpressed in STAT1 Ko-L929 cells, *Ifn-β*(M) and *Rig-i* (N) mRNA were detected by qPCR. The data represent the means ± SD, from three independent experiments. *$p < 0.05$, **$p < 0.01$, ***$p < 0.001$ using Student $t$ test. This data underlying this Figure can be found in S1 Data and S1 Raw Images.

In summary, USP39 coordinates innate immune signaling through dual regulation, bolstering antiviral immunity by promoting RIG-I pre-mRNA splicing and STING deubiquitination and stabilization, synchronizing anti-RNA and DNA antiviral signaling pathways to facilitate type I interferons expression (Fig 8). Elucidating these roles highlights USP39 plays a dominant role in immune regulation, offering potential therapeutic target for viral infections.

## Discussion

The global COVID-19 pandemic has inflicted catastrophic morbidity and socioeconomic devastation, starkly exposing critical knowledge gaps in virological pathogenesis and therapeutic vulnerabilities, thereby underscoring the exigency for paradigm-shifting advancements in antiviral countermeasure development and host-pathogen interaction elucidation. USP39 is a deubiquitinating enzyme that exists ubiquitously in cells and is involved in many important cellular activities. Most research into USP39 function and mechanism of action have focused on cancers, including hepatocellular carcinoma [35], ovarian cancer [37], breast cancer [38], and colon cancer [43], or on DNA damage and repair [34]. Nevertheless, our understanding on the role of USP39 has been limited due to the original inference that this protein lacked deubiquitination activity because of it lacked the characteristic sequence in its DUB domain [44,45]. Research now supports that USP39 can not only splice pre-mRNAs for their maturation [37,39] but also deubiquitinates target proteins including ZEB1, SP1, FOXM1, and STAT1 [35,36,38,40]. A few reports, however, have implicated USP39 in the regulation of antiviral infection [40]. It seems that USP39 has been implicated as a potential regulator, but the role and mechanism are unclear.

Here, we confirmed that USP39-mediated splicing of *Rig-i* pre-mRNA and deubiquitination of STING protein positively regulates the antiviral immune response in the context of RNA and DNA viral infections, respectively. USP39 is an important component of the splicing complex [45], but support for USP39-mediated mRNA and protein regulation (via splicing and ubiquitination, respectively) in the antiviral immune response was minimal prior to this study. RIG-I is ubiquitously expressed and upon encountering intracellular viral RNA, binds and exposes CARD to recruit downstream signaling molecules to activate immune response [46,47]. The intact tandem CARD domains of RIG-I are crucial for its function. Its splice variant (SV), which harbors a short deletion of amino acids 36–80 within the first CARD domain, lacks the ability to bind TRIM25, the capacity for CARD domain ubiquitination, and downstream signal transduction capability [48]. Previous studies showed that the E3 ubiquitin ligase TRIM25 robustly induces the K63-linked ubiquitination of RIG-I, thus regulating the RIG-I signaling pathway during the antiviral innate immune response [49]. XRCC4 also enhances RIG-I oligomerization and ubiquitination, thus decreasing the replication of viral RNA in host cells [50]. Yet despite much has been reported about the deubiquitination regulation of RIG-I on antiviral immune response, little know about the splicing pre-mRNA of RIG-I. In a previous study, we confirmed that USP39 induces ADAM9 mRNA splicing maturation to promote human glioma cell migration and invasion [51]. In this study, we demonstrate that USP39 promotes *Rig-i* pre-mRNA maturation. This splicing activity also can be performed by USP39-CA, which renders USP39 unable to elicit its deubiquitinating activity. Notably, our data have validated that USP39 modulates the pre-mRNA splicing of RIG-I. As a core component of the U4/U6-U5 tri-snRNP spliceosome

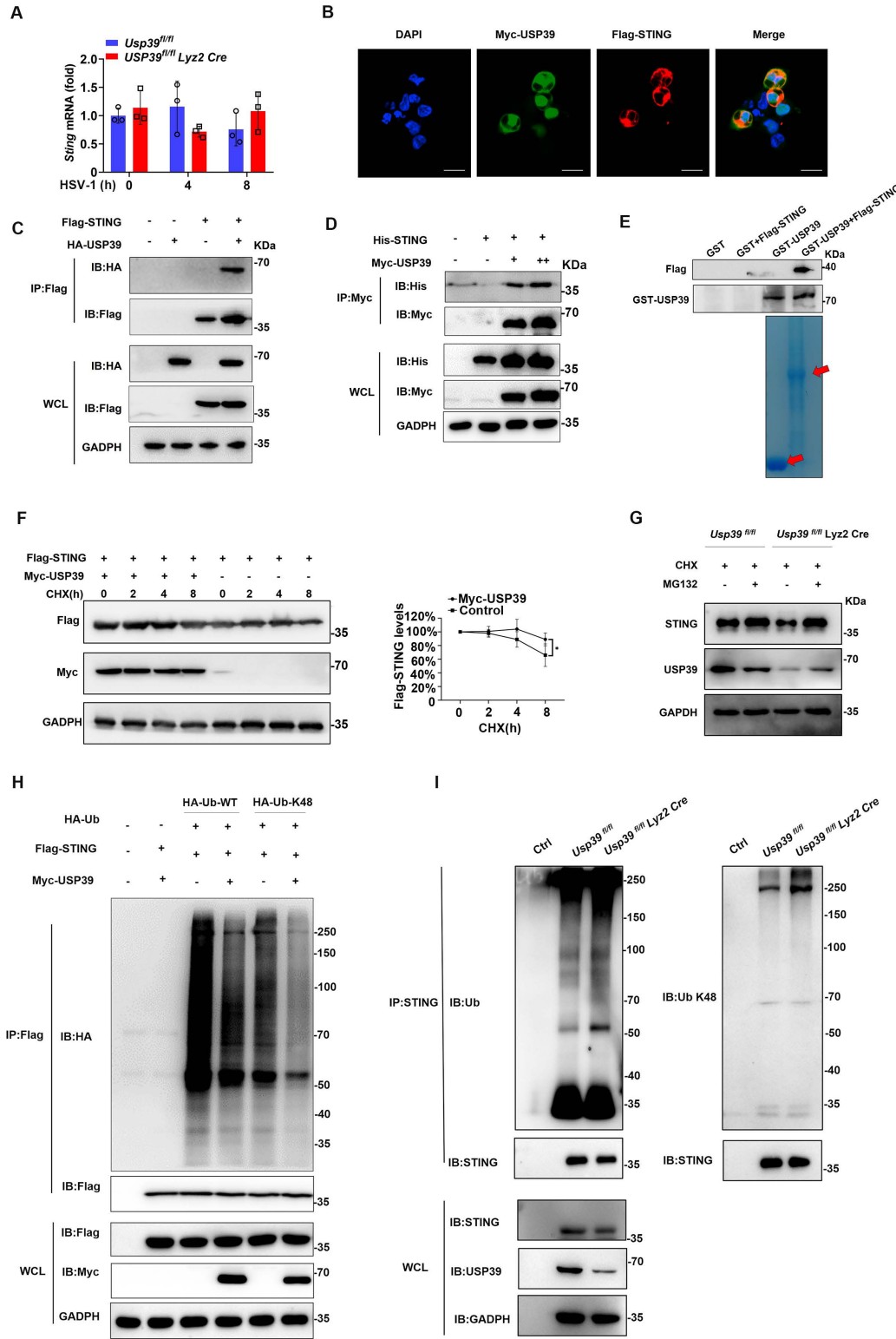

**Fig 6. USP39 mediated K48-linked deubiquitination stabilizes STING. (A)** STING mRNA was measured by qPCR in *Usp39*fl/fl and *Usp39*fl/fl *Lyz2 Cre* macrophages infected with HSV-1 for the indicated time. **(B)** Flag-STING and Myc-USP39 were co-overexpressed in HEK293T cells, and their

co-localization was detected by confocal microscopy. The cells were stained with anti-Flag and anti-Myc antibodies; the nuclei were stained with DAPI. **(C, D)** Flag-STING and HA-USP39 (C) or His-STING and Myc-USP39 (D) were co-overexpressed in HEK293T cells before immunoprecipitation with anti-Flag M2 magnetic beads or anti-His antibodies. The cell lysates were analyzed by western blotting. **(E)** GST, GST-USP39, and in vitro translated Flag-STING were analyzed by GST pulldown assay. **(F)** Flag-STING and Myc-USP39 were co-overexpressed in HEK293T cells before the cells were exposed to CHX (100 μg/ml) for the indicated time. The cell lysates were analyzed by western blotting. **(G)** *Usp39*fl/fl and *Usp39*fl/fl *Lyz2 Cre* macrophages were treated with CHX (100 μg/ml) and then treated with MG132 (20μM) for 6 h. STING expression in the cell lysates was analyzed by western blotting. **(H)** Flag-STING, HA-Ub-WT, HA-Ub-K48, and Myc-USP39 were co-overexpressed in HEK293T cells, before the cells were exposed to MG132 (20 μM) for 6 h. The cell lysates were precipitated with anti-Flag M2 magnetic beads and STING ubiquitination was determined by western blotting. **(I)** *Usp39*fl/fl and *Usp39*fl/fl *Lyz2 Cre* macrophages were treated with MG132 (20 μM) for 6 h before the cell lysates were precipitated with anti-STING antibodies, then ubiquitination of wild type and K48 was detected by western blotting. The data represent the means ± SD, from three independent experiments. *$p < 0.05$, **$p < 0.01$, ***$p < 0.001$ using Student *t* test. This da*ta underlying this Figure can be found in S1 Data and S1 Raw Images.

complex, USP39 interacts with key splicing regulators including EFTUD2, PRPF3, SART1, and PRP8 to govern spliceosome assembly and catalytic activation [52,53]. Beyond its established role in intron excision, the robust enrichment of RIG-I mRNA in USP39 RIP assays implies that USP39 may also contribute to bind to the mature RIG-I mRNA and maintain the stability of the RIG-I mRNA. We propose that USP39 regulates RIG-I expression not only bind to the RIG-I pre-mRNA to promote RIG-I mRNA maturation, but also bind to the mature RIG-I mRNA to stabilize it. This multi-layered control at the post-transcriptional level, rather than direct protein modification or degradation, likely underlies how USP39 regulates RIG-I expression and downstream antiviral signaling.

Other studies have revealed that STING performs its anti-DNA viral function through protein post-translational modifications or stabilization. For example, STING is negatively regulated by USP13 during HSV-1 infection [54]. OTUD5 stabilizes STING by deubiquitinating function, which also helps to defend against DNA viral infection [55]. USP22 regulates STING protein levels and type III IFN signaling to protect against SARS-CoV-2 infection [56]. USP21 or OTUB1 also exert a significant effect on the innate immune response to viruses [57,58]. In this work, USP39 but not USP39-CA, however, can stabilize STING protein, indicating that differing from RIG-I, STING is dependent on the deubiquitinating activity of USP39. Furthermore, although we have performed numerous experiments to demonstrate that USP39 can stabilize STING through deubiquitination, we are currently unable to fully prove that USP39 can only exert its effect by removing the covalently linked ubiquitin chains of STING. Further research is needed to confirm this.

Moreover, we saw that in vivo, a USP39 deficiency impairs the murine defense against RNA or DNA viral infection. Mechanistically, *Ifn-β* and *Ifn-α4* mRNA levels were decreased in various organs in our model mice. Indeed, we saw that lung inflammation following VSV infection was more notable in *Usp39*fl/fl *Lyz2 Cre* mice, and the survival rate was decreasing. Recovery of USP39 levels in *Usp39*fl/fl *Lyz2 Cre* BMDMs could, however, rescue the weak *Ifn-β* response to VSV or HSV-1 infection. While our findings regarding USP39 have been validated in vitro and in vivo in mouse models, clinical research will be warranted alongside further mechanistic analyses of USP39 activity.

USP39 is a central role to regulate inflammatory response. Results of our previous study showed that USP39-deficient macrophages were unable to stabilize basal IκBα levels, causing over-stimulation of the inflammatory response after LPS or *E.coli* exposure [31]. Viral infections also trigger severe inflammatory diseases. For example, SARS-CoV-2 can trigger devastating immune dysregulation caused by a cytokine storm that leads to systemic inflammation and, in severe cases, multiorgan dysfunction [59]. The hepatitis A and B viruses can trigger fulminant viral hepatitis, in which an inflammatory infiltrate is produced alongside high FGL2, IFNγ, IL-18, and IL-1β expression [60]. RNA or DNA viral infections also activate the NF-κB signaling pathway [61]; in line with this, we saw that *Il-6* expression (inflammatory marker) increased in *Usp39*fl/fl *Lyz2 Cre* macrophages infected with VSV or HSV-1. We also saw higher replication of VSV and HSV-1, and the lower expression of *Isgs*, *Ifn-β*, and *Ifn-α4* in *Usp39*fl/fl *Lyz2 Cre* macrophages. These data suggest that USP39 plays a dual role in antiviral immunity by regulating type I IFNs and inflammatory response to protect the host in viral infection.

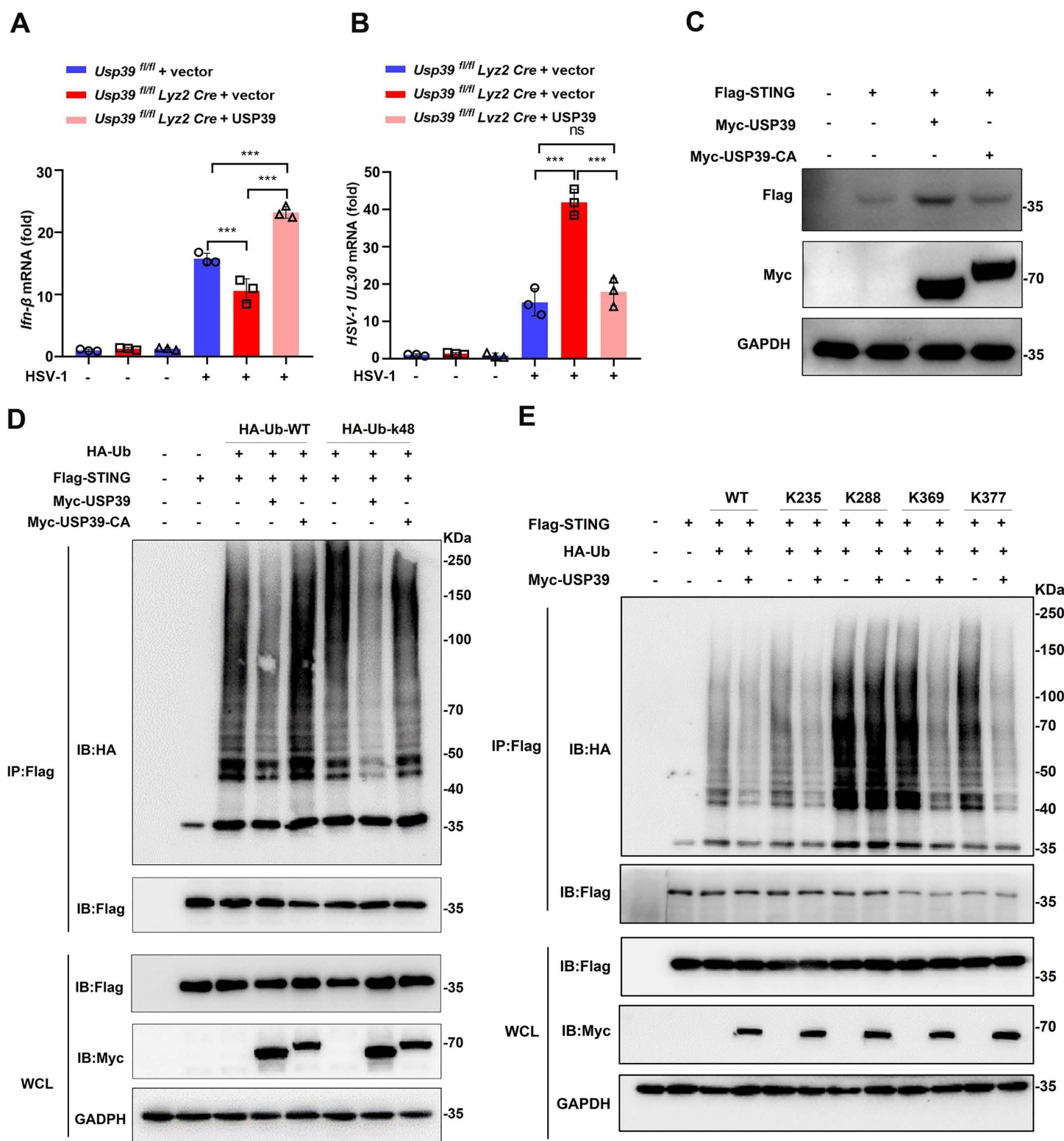

**Fig 7. USP39-mediated K48-linked deubiquitination of STING at Lys288 depends on its deubiquitinating enzyme activity. (A, B)** USP39 was overexpressed using a lentiviral in *Usp39*fl/fl and *Usp39*fl/fl *Lyz2 Cre* BMDMs before the cells were infected with HSV-1 (MOI = 10) for 12h. *Ifn-β* (A) and *HSV-1 UL30* (B) mRNA levels were measured by qPCR. **(C)** A control vector, Myc-USP39 or Myc-USP39- CA were overexpressed in HEK293T cells before the cell

lysates were analyzed by western blotting. **(D)** Flag-STING, HA-Ub-WT, HA-Ub-K48, Myc-USP39, Myc-USP39-CA were co-overexpressed in HEK293T cells before the cells were treated with MG132 (20 μM) for 6 h. The cell lysates were precipitated with anti-Flag M2 magnetic beads and STING ubiquitination was determined by western blotting. **(E)** Flag-STING, Flag-STING K235, Flag-STING K288, Flag-STING K369, Flag-STING K377, HA-Ub-WT, Myc-USP39 were co-overexpressed in HEK293T cells before the cells were treated with MG132 (20 μM) for 6 h. The cell lysates were precipitated with anti-Flag M2 magnetic beads and STING ubiquitination was determined by western blotting. The data represent the means ± SD, from three independent experiments. *$p < 0.05$, **$p < 0.01$, ***$p < 0.001$ using Student $t$ test. This da*ta* underlying this Figure can be found in S1 Data and S1 Raw Images.

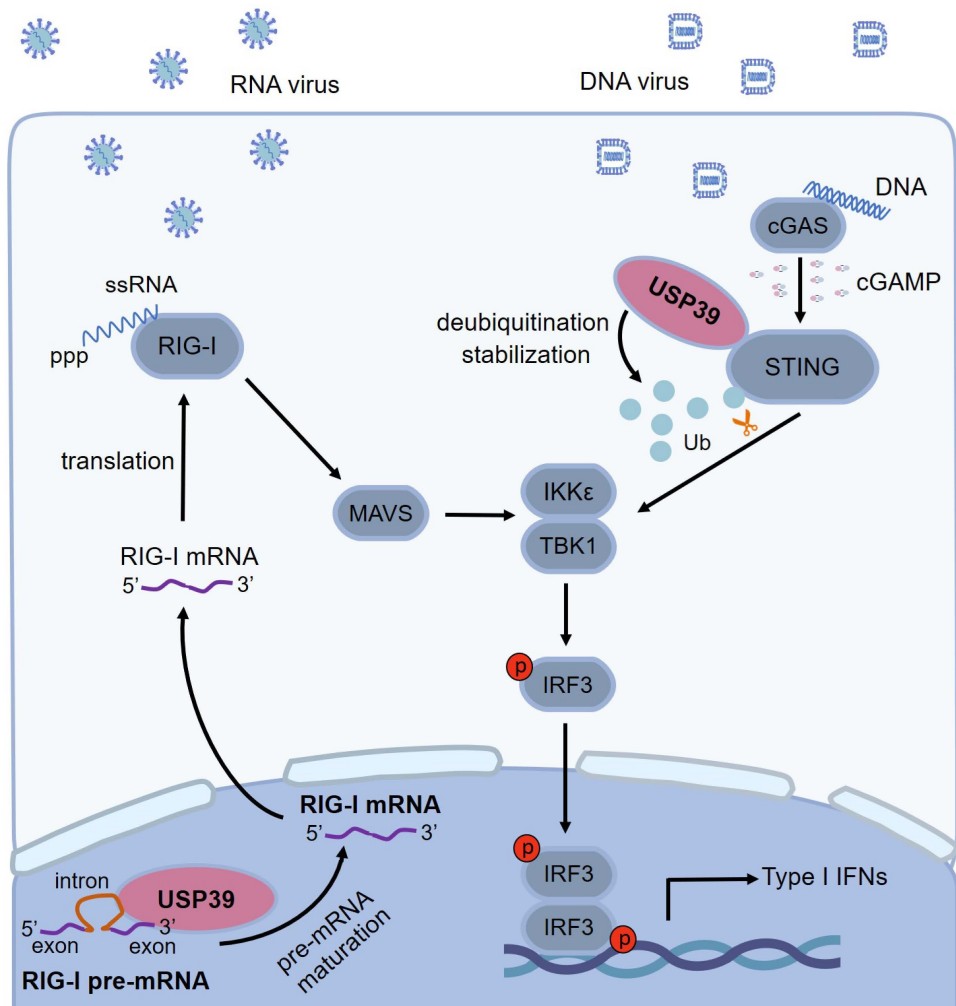

**Fig 8. The dual role of USP39 in regulating RIG-I and STING to initiate an immune response to RNA and DNA viral infections.** RNA and DNA viruses stimulate the RIG-I and cGAS-STING signaling pathways, respectively. USP39 splices RIG-I pre-mRNA and deubiquitinates K48-linked STING, at K288 to regulate IFN-β, IFN-α4, and ISG expression, promoting cellular antiviral responses in innate immunity.

In summary, we provide physiological and biochemical evidence that USP39 regulates RNA and DNA viral infections by targeting RIG-I and STING signaling, which offers insight into the previously unappreciated immune regulatory role of USP39. The positive antivirus role and the anti-inflammatory function in response to a virus infection leads us to believe that USP39 is essential to host antiviral processes. We consider, therefore, that USP39 has potential to serve as a point of intervention for various RNA and DNA viruses.

## Materials and methods

### Ethics statement

Mice were maintained under specific-pathogen-free conditions and housed in the Laboratory Animal Center of Zhejiang University and Animal Center of Shenzhen University, China. All experiments using mice were conducted in accordance with the Institutional Animal Care and Use Committee, and the experimental protocols were approved by Laboratory Animal Welfare and Ethics Committee of Zhejiang University (IACUC ZJU20160015).

### Mice, cells, and virus

*Usp39*$^{fl/fl}$ mice and *Usp39*$^{fl/fl}$ *Lyz2 Cre* mice were acquired and bred according to a previous study [31]. The *Usp39*$^{fl/fl}$ *Lyz2 Cre* mice did not express *Usp39* in myeloid cells specifically, and *Usp39*$^{fl/fl}$ littermate mice were used as controls.

All cells were cultured at 37 °C with 5% $CO_2$. HEK293T cells and HeLa cells were derived from the Culture Collection of the Chinese Academy of Sciences (Shanghai, China) and were cultured in DMEM medium with 10% FBS and 1% penicillin-streptomycin. Primary peritoneal macrophages were isolated from 8- to 12-week-old mice through intraperitoneal injection into 2 ml of 3% thioglycollate broth and then cells cultured in RPMI-1640 (10% FBS and 1% penicillin-streptomycin). Bone-marrow-derived macrophages (BMDMs) were obtained from the bone marrow of mice and differentiated in RPMI-1640 (10% FBS and 1% penicillin-streptomycin) with 100 ng/ml M-CSF (recombinant mouse macrophage colony-stimulating factor) for 1 week. The USP39 lentivirus was purchased from viGene Biosciences (USA). VSV, H1N1 PR8, HSV-1, and HSV-2 were acquired from the Culture Collection of the Chinese Academy of Sciences (Shanghai, China).

### Reagents

RLR, cGAS-STING Pathway Sampler Kit, and ubiquitin antibody were purchased from Cell Signaling Technology (Beverly, MA, USA). USP39 antibodies were purchased from Bethyl (Texas, USA), GAPDH, HA-Tag, His-Tag and Myc-Tag were purchased from Proteintech (Wuhan, China), and Flag-Tag antibodies were purchased from Genscript (Nanjing, China). Anti-Flag M2 magnetic beads were purchased from Sigma (USA). Anti-Myc magnetic beads were purchased from Bimake (USA). Cycloheximide (CHX), Carbobenzoxy-Leu-Leu-leucinal (MG132), and chloroquine (CQ) were purchased from Selleck Chemicals (USA). dsDNA, cGAMP, and low molecular weight polyinosine-polycytidylic acid [Poly (I:C)] were purchased from Invivogen (France). The ELISA kit for mouse IFN-β was purchased from MULTI SCIENCES (Cat: EK2236-96; Hangzhou, China).

### Plasmids and RNA interference

The JetPrime transfection reagent (Polyplus Transfection, New York, NY, USA) was used to transfect plasmids into HEK293T cells. Transfected cells were cultured at 37 °C with 5% $CO_2$ overnight, and then treated or collected for further experiments. The Myc-USP39 C306A mutation (Myc-USP39-CA, enzyme-inactivating mutation) was generated through site-specific mutagenesis, as previously described [62]. Macrophages ($1 \times 10^6$ cells) were transfected with *Usp39*-small interfering RNA (siRNA) (20 nmol/ml) and incubated for 36 h before further use. The mouse *Usp39* siRNA sequence was as follows: 5′-CAACGACUAUGCAAAUGCUTT-3′.

### Quantitative PCR and ELISA

Total cell RNA was extracted in Trizol reagent (Takara, Japan). A CFX96 Touch Real-Time PCR System (Bio-Rad, Hercules, CA, USA) was used for qPCR, with Hieff qPCR SYBR Green Master Mix (Yeasen, Shanghai, China), according to the manufacturer's instructions. IFN-β protein levels were analyzed in cell supernatants or mouse sera by ELISA, according to the manufacturer's instructions. Murine *Ifn-β* and *Ifn-α4* and human *Ifn-β* primers were designed as previously described [63]. Other primer sequences are detailed in S1 Table.

## TCID50 assay, flow cytometric assay, and immunofluorescence assay

Cells were infected with VSV (MOI = 1) for 12 or 24 h, and the supernatants were analyzed by TCID50 according to a previous study [63]. Flow cytometric assays were performed according to the guidelines. Cells were infected with PBS or VSV-eGFP (MOI = 1) for 12 h and then digested with pancreatin and washed with PBS. The results were analyzed by Flowjo. Cells were cultured in a 12-well plate, infected with VSV-eGFP (MOI = 1) for 12 h, washed with PBS, and detected by fluorescence microscopy.

## Western blotting

HEK293T cells and macrophages were lysed in NP40 lysis buffer containing 1× protease inhibitor mix and incubated on ice for 40 min. Then, the samples were centrifuged at 12,000$g$ for 15 min. The collected supernatants were added to 1× loading buffer and boiled for 10 min. Equal protein mixtures were loaded and separated on 8%–10% SDS-PAGE gels, then transferred to PVDF membranes. The membranes were incubated in 5% skim milk at room temperature for 1 h and then incubated with the indicated antibodies overnight at 4 °C. After washing, the membranes were incubated with a secondary antibody for 1.5 h at room temperature. Proteins were visualized using Chemiluminescent reagent kits (Thermo Fisher Scientific, Waltham, MA, USA) and detected by FluorChem E (Cell Biosciences, USA).

## Ubiquitination assay

HEK293T cells and macrophages were lysed in IP buffer (containing 1% of SDS and a 1× protease inhibitor mixture) and boiled for 5 min at 95 °C. Supernatant was collected and diluted 10-fold in IP buffer, following IP with indicated antibody for 2 hours, and then incubated with protein A/G Plus-Agarose (Bimake, TX, USA) or Anti-Flag M2 magnetic beads at 4 °C for 12 h. The samples were then washed in immunoprecipitation buffer five times, and the beads were diluted in SDS buffer and boiled for 10 min. Equal amounts of protein were analyzed by western blotting, as described previously [64].

## In vitro GST-pull down assay

GST only and full-length GST-USP39 plasmids were transformed into BL21 cells, and were induced with IPTG (1 mM) at 16 °C for 20 h. The BL21 cells were lysed in lysis buffer (50 mM Na2HPO4, 300 mM NaCl, 10 mM imidazole) and the proteins were purified on a glutathione Sepharose 4B matrix (Sigma), before being collected in glutathione elution buffer (20 mM glutathione in 500 mM Tris–HCl (1 M, PH 8.0)). Flag-STING proteins were expressed in HEK293T cells, and STING was pulled down using anti-Flag M2 magnetic beads. Purified GST and GST-USP39 were incubated with Flag-STING at 4 °C overnight, and then incubated with a glutathione Sepharose 4B matrix at 4 °C for 2 h to pull down the target protein.

## RNA-binding protein immunoprecipitation assay

RIP analysis was conducted using a Magna RIP KIT (Millipore, Billerica, MA, USA) according to the manufacturer's protocol. Co-precipitated RNAs were isolated, subjected to PCR analysis and qPCR analysis as described [51].

## Virus infection mice model

*Usp39*[fl/fl] and *Usp39*[fl/fl] *Lyz2 Cre* 8–12-week-old mice were exposed to VSV via tail vein injection (250 μl, $1 \times 10^9$/g) or HSV-1 via tail vein (250μl, $1 \times 10^8$/g) or intraperitoneal injection (1 ml, $1 \times 10^8$/g). Murine sera were collected and detected by ELISA. Spleen, liver, and lung tissues were removed, and the RNA was extracted and analyzed by qPCR, and supernatants of equal weighting tissue lysate were analyzed by TCID50. Lungs were also dissected, fixed, and stained with hematoxylin-eosin using standard procedures, and histological changes were detected under a light microscope. *Usp39*[fl/fl]

and *Usp39^{fl/fl} Lyz2 Cre* 8–12-week-old mice were exposed to VSV via tail vein injection (350 μl, $1 \times 10^9$/g), and the survival of the mice then recorded. All *Usp39^{fl/fl}* and *Usp39^{fl/fl} Lyz2 Cre* mice were age- and sex-matched.

## Lentivirus-mediated USP39 transfection

*Usp39^{fl/fl}* and *Usp39^{fl/fl} Lyz2 Cre* BMDMs were generated and cultured in RPMI-1640 (10% FBS and 1% penicillin-streptomycin) with 100 ng ml$^{-1}$ M-CSF (recombinant mouse macrophage colony-stimulating factor) for 1 week. On the second day of culture, the *Usp39^{fl/fl}* and *Usp39^{fl/fl} Lyz2 Cre* BMDMs were infected with a USP39 lentiviral vector or a control vector (MOI = 10).

## Statistical analysis

All data are presented as the means ± SD of at least three independent experiments. Student *t* test is to test for statistical differences between two groups, respectively. Mouse survival was determined from Kaplan–Meier survival curves. $p < 0.05$ was considered to indicate a statistically significant difference.

## Supporting information

**S1 Fig. *Usp39^{fl/fl} Lyz2 Cre* macrophages showed higher inflammatory response and lower IFN-stimulated genes expression after infected with virus or transfected with cGAMP. (A)** The efficiency of HA-USP39 overexpression in HEK293T cells. **(B)** HeLa cells overexpressed a control vector or HA-USP39, and then were infected with VSV-eGFP (MOI = 1) for 12 h before examination by fluorescence microscopy. Scale bar = 100 μm. **(C)** HeLa cells overexpressed a control vector or HA-USP39, and then were infected with H1N1 PR8 (MOI = 1) for 12 h before *Ifn-β* mRNA levels were measured by qPCR. **(D, E)** *Usp39^{fl/fl}* and *Usp39^{fl/fl} Lyz2 Cre* macrophages were infected with VSV (MOI = 1) (D) or HSV-1 (MOI = 10) (E) for the indicated time before *Il-6* mRNA levels were measured by qPCR. **(F)** *Usp39^{fl/fl}* and *Usp39^{fl/fl} Lyz2 Cre* macrophages were transfected with cGAMP (1 μg/mL) for the indicated time, before *Isg15, Isg56*, and *Cxcl10* mRNA levels were measured by qPCR. **(G)** *Usp39* was knocked down in macrophages, and *Usp39* mRNA levels were measured by qPCR. **(H, I)** Flag-MAVS and Myc-USP39 (H) or Myc-USP39 (I) were overexpressed in HEK293T cells. The cell lysates were analyzed by western blotting, and *MAVS* mRNA was detected by qPCR. The data represent the means ± SD, from three independent experiments. *$p < 0.05$, **$p < 0.01$, ***$p < 0.001$ using Student *t* test. This data underlying this Figure can be found in S1 Data and S1 Raw Images.
(TIF)

**S2 Fig. USP39 regulates RIG-I protein through RIG-I mRNA maturation after infected with VSV or transfected with Poly (I:C). (A, B)** The region from exon 3 to 4 in the human *Rig-i* primary transcript (A) and exon 2 to 3 in the mouse *Rig-i* primary transcript (B) was used to design specific primers to detect spliced and unspliced *Rig-i* mRNA. **(C, D)** Special *Rig-i* mRNA levels were measured by qPCR in *Usp39^{fl/fl}* and *Usp39^{fl/fl} Lyz2 Cre* macrophages after they had been transfected with Poly (I:C) (1 μg/mL) at the indicated time (C). A control vector or HA-USP39 were overexpressed in HeLa cells before being infected with VSV (MOI = 1) at the indicated time. Special *Rig-i* mRNA levels were measured by qPCR (D). **(E)** All the 18 exons/introns in BMDM were detected by qPCR. **(F)** *Stat1* mRNA in *Stat1* Ko-L929 cell were detected by qPCR. **(G)** HA-USP39 overexpressed in STAT1 Ko-L929 cell, and the cell lysates were analyzed by western blotting. (H) Myc-RIG-I and HA-USP39 were co-overexpressed in HEK293T cells and performed by RIP, *Rig-i* mRNA levels measured by qPCR. (I) Flag-STING, HA-Ub, Myc-OTUB1, Myc-USP39 were co-overexpressed in HEK293T cells before the cells were treated with MG132 (20 μM) for 6 h. The cell lysates were precipitated with anti-HA magnetic beads, and Flag-STING protein was determined by western blotting. The data represent the means ± SD, from three independent experiments. *$p < 0.05$, **$p < 0.01$, ***$p < 0.001$ using Student *t* test. This data underlying this Figure can be found in S1 Data and S1 Raw Images.
(TIF)

**S3 Fig. USP39 does not influence STING mRNA level but deubiquitinates STING at K288. (A)** Conserved lysine sites in different species.
(TIF)

**S1 Table. Primers used in this study.**
(DOCX)

**S1 Data. Raw numerical data are included in this study.**
(XLSX)

**S1 Raw Images. Image-based raw data are included in this study.**
(PDF)

## Acknowledgments

We would like to acknowledge the support provided by Instrument Analysis Center of Shenzhen University for the assistance with confocal microscopy analysis. We thank Prof. Xuetao Cao (Peking Union Medical College) for kindly providing *Stat1*-knockout L929 cell. We thank Jessica Kate Tamanini (Scientific Editor, Shenzhen University Medical School) to perform the language polishing on the revised version.

## Author contributions

**Conceptualization:** Jiazheng Quan, Xibao Zhao, Weilin Chen.

**Data curation:** Jiazheng Quan.

**Formal analysis:** Jiazheng Quan, Xibao Zhao, Wei Chen, Han Wu.

**Funding acquisition:** Xibao Zhao, Yue Xiao, Weilin Chen.

**Investigation:** Jiazheng Quan, Xibao Zhao, Shaoying Chen, Wei Chen, Qianqian Di, Xunwei Li, Jiajing Zhao, Yue Xiao.

**Methodology:** Jiazheng Quan, Shaoying Chen, Hongrui Li, Yue Xiao.

**Resources:** Hongrui Li, Qianqian Di, Zherui Wu.

**Supervision:** Weilin Chen.

**Validation:** Jiazheng Quan, Xunwei Li, Jiajing Zhao, Han Wu, Jin Chen.

**Writing – original draft:** Jiazheng Quan, Xibao Zhao, Weilin Chen.

**Writing – review & editing:** Jiazheng Quan, Xibao Zhao, Weilin Chen.

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
