## [Editor Report · Decision Letter 0]

12 May 2025

Dear Dr chen,

Thank you for submitting your manuscript entitled "USP39 augments antiviral immunity through dual mechanisms of promoting RIG-I pre-mRNA splicing and STING deubiquitination" for consideration as a Research Article by PLOS Biology.

Your manuscript has now been evaluated by the PLOS Biology editorial staff, as well as by an academic editor with relevant expertise, and I am writing to let you know that we would like to send your submission out for external peer review.

Once your full submission is complete, your paper will undergo a series of checks in preparation for peer review. After your manuscript has passed the checks it will be sent out for review. To provide the metadata for your submission, please Login to Editorial Manager (https://www.editorialmanager.com/pbiology) within two working days, i.e. by May 14 2025 11:59PM.

Kind regards,

Melissa

Melissa Vazquez Hernandez, Ph.D.

Associate Editor

PLOS Biology

---

## [Decision Letter · Decision Letter 1]

12 Jun 2025

Dear Dr chen,

Thank you for your patience while your manuscript "USP39 augments antiviral immunity through dual mechanisms of promoting RIG-I pre-mRNA splicing and STING deubiquitination" was peer-reviewed at PLOS Biology. Your manuscript has been evaluated by the PLOS Biology editors, an Academic Editor with relevant expertise, and by two independent reviewers.

As you will see in the reviewer reports, which can be found at the end of this email, although the reviewers find the work potentially interesting, they have also raised a substantial number of important concerns. Based on their specific comments and following discussion with the Academic Editor, it is clear that a substantial amount of work would be required to meet the criteria for publication in PLOS Biology. However, given our and the reviewer interest in your study, we would be open to inviting a comprehensive revision of the study that thoroughly addresses all the reviewers' comments Reviewer 1 thinks that the study is of high interest but, while the first half of the manuscript is good, the second part is not well supported. This reviewer gives several suggestions to enhance the strength of the data presented from Fig 3 onwards, such as also analysing other exons and introns from RIG-I. S/He also raised concerns regarding overstatements and technical issues in Figs 6 and 7. Reviewer 2 has similar thoughts as Reviewer 1, and thinks the mechanistic conclusions must be better supported. This reviewer suggests that you evaluate alternative pathways of regulation by USP39 and how the re-expression affects RIG-I splicing. S/He requests that you repeat the ubiquitination experiments and to test ubiquitination in different STING mutants. Please be aware that both reviewers mentioned that the language should be improved. The revision must address all reviewers' concerns.

Given the extent of revision that would be needed, we cannot make a decision about publication until we have seen the revised manuscript and your response to the reviewers' comments. Your revised manuscript would need to be seen by the reviewers again, but please note that we would not engage them unless their main concerns have been addressed.

We appreciate that these requests represent a great deal of extra work, and we are willing to relax our standard revision time to allow you 6 months to revise your study. Please email us (plosbiology@plos.org) if you have any questions or concerns, or envision needing a (short) extension.

**IMPORTANT - SUBMITTING YOUR REVISION**

*Resubmission Checklist*

*Published Peer Review*

*PLOS Data Policy*

*Blot and Gel Data Policy*

Sincerely,

Melissa

Melissa Vazquez Hernandez, Ph.D.

Associate Editor

PLOS Biology

REVIEWERS' COMMENTS:

Reviewer #1:

The RIG-I/MAVS and cGAS/STING pathways play crucial roles in innate immune defence against viral pathogens but can also cause or contribute to detrimental inflammation. Many regulatory mechanisms have evolved in both pathways. This manuscript adds to the field's growing understanding of such mechanisms. Figs 1-3 nicely show in different in vitro and in vivo settings that USP39 positively controls innate immunity, particularly type I interferon induction, during RNA and DNA virus infection. The second part of the manuscript then dissects the underlying mechanisms. The authors claim that USP39 regulates RIG-I pre-mRNA splicing and STING ubiquitination. This is potentially of high interest because the same protein (USP39) controls different innate immune sensing pathways using very different molecular mechanism. This insight is not only of fundamental importance to our understanding of how innate immunity is controlled but potentially also relevant to targeting RIG-I and STING signalling for therapy. However, in my view, the data in Figs 4-7 are unconvincing and do not support the mechanistic claims well. I hope the authors will find my suggestions useful in further testing their model.

Major points

1. Fig 3, HSV-1 infection: did the authors analyse viral replication, survival and/or pathology (as presented for VSV infections)?

2. Fig 4:

a. In Fig 4a and b, it appears MAVS is expressed at lower levels in USP39-deficient cells. Did the authors observe this effect reproducibly, and if so, how does this impact the interpretation of the results?

b. The experiments shown in Fig 4c and h would benefit from including cells that endogenously express RIG-I and STING. In addition, a negative control is missing: please include another Flag-tagged protein that is unaffected by USP39 overexpression.

3. Fig 5 and S2 (RIG-I splicing):

a. Fig S2a,b: For clarity, please indicate (using arrows) the positions of the primers used for RT-qPCR.

b. The mouse RIG-I gene has 18 exons. Why did the authors choose to study exons 2-3 only? Other exons and introns should be included to strengthen the conclusions.

c. Fig 5h / lines 245-250. How was RIG-I overexpressed in this setting? Did the authors transfect a RIG-I encoding plasmid? If yes, did this plasmid include the entire RIG-I gene or only the RIG-I cDNA? I could not find information in the methods regarding this point. Assuming a RIG-I cDNA plasmid was used here, the data are misinterpreted and would only show USP39 binding to mature RIG-I mRNA. However, the authors conclude that USP39 binds the pre-mRNA and allows mRNA maturation. Please clarify. Please also provide protein / western blot data to validate the IP and explain why both PCR and qPCR were used here.

d. Line 260/261: these conclusions appear overstated. An alternative scenario is that USP39 has an impact of RIG-I pre-mRNA and/or mature mRNA stability.

e. Have the authors attempted northern blotting to demonstrate defects in RIG-I pre-mRNA processing using an alternative method (not involving amplification)?

f. Line 348. The authors must cite and discuss this article on a RIG-I splice variant: pmid: 18948594.

4. There are a number of technical concerns regarding Figs 6 and 7 (STING deubiquitination):

a. Fig 6c should include an extra negative control: expression of Myc-USP39 alone, followed by anti-FLAG immunoprecipitation.

b. Fig 6e should include an extra negative control: pulldown using GST and Flag-STING to exclude non-specific binding of STING to the beads.

c. Fig 6f: comparing lanes 3 and 4, there is a strong reduction in the Flag-STING signal; however, this is not reflected in the quantification below the blots. Moreover, for the quantification, the data from the three independent repeats need to be shown together.

d. Fig 6g should include an extra control: MG132 treatment of USP39-sufficient cells to exclude non-specific binding of MG132.

e. Fig 7d: The authors conclude that the catalytically dead USP39 is not able to de-ubiquitinate STING. However, comparing the HA signal in Flag-IPs (lanes 4 vs 5 and lanes 7 vs 8), differences appear marginal. Have the authors quantified results from multiple independent experiments?

5. PMID 33127822 suggests that USP39 regulates STAT1. The authors should exclude that the effects observed here are an indirect consequence of USP39 facilitating IFN-IFNAR-JAK/STAT signalling, perhaps at baseline. In particular, the RIG-I gene is IFN-inducible and controlled by STAT1. Key experiments should be repeated in STAT1 KO cells.

6. Many western blots lack size markers. Please add these to all western blots shown in main and supplementary figures.

7. The manuscript requires substantial English language editing.

Minor points

1. Line 27: RIG-I is not an adaptor. It is an RNA sensor. The adaptor protein in the pathway is MAVS.

2. References 10, 16, 19, 27 and 36 are very old. Please replace with more recent reviews.

3. Line 81: If citing ref 18 (describing the discovery of MAVS), please also cite the three other papers thar simultaneously discovered MAVS:

a. Seth, R.B., Sun, L., Ea, C.K., and Chen, Z.J. (2005). Identification and characterization of MAVS, a mitochondrial antiviral signaling protein that activates NF-kappaB and IRF 3. Cell 122, 669-682. 10.1016/j.cell.2005.08.012.

b. Meylan, E., Curran, J., Hofmann, K., Moradpour, D., Binder, M., Bartenschlager, R., and Tschopp, J. (2005). Cardif is an adaptor protein in the RIG-I antiviral pathway and is targeted by hepatitis C virus. Nature 437, 1167-1172. 10.1038/nature04193.

c. Xu, L.G., Wang, Y.Y., Han, K.J., Li, L.Y., Zhai, Z., and Shu, H.B. (2005). VISA is an adapter protein required for virus-triggered IFN-beta signaling. Molecular cell 19, 727-740. 10.1016/j.molcel.2005.08.014.

4. Line 88. Alongside ref 28, this paper also needs to be cited here: Wu, J., Sun, L., Chen, X., Du, F., Shi, H., Chen, C., and Chen, Z.J. (2013). Cyclic GMP-AMP is an endogenous second messenger in innate immune signaling by cytosolic DNA. Science 339, 826-830. 10.1126/science.1229963.

5. Line 137: the term "immunofluorescence" appears to be incorret here, unless the GFP signal was detected using an anti-GFP antibody.

6. The p-TBK1 blot in Fig 4d is very dark. Can the authors provide a shorter exposure?

7. Lines 230-232: I suggest replacing "was also downregulated" with "showed reduced induction" to better represent the data: USP39-deficient cells still show an increase in RIG-I mRNA levels after infection or pI:C transfection.

8. Please replace "Extron" with "Exon" in the text and figures.

9. Line 487. Please include a reference to the publication where these steps were described.

10. Line 806: typo (STIGN)

Reviewer #2:

This manuscript by Quan et al. reports that USP39 promotes anti-viral responses via RIG-I and cGAS-STING to suppress viral replication in cultured cells (murine macrophages and HeLa cells) and in vivo. Using Usp39fl/fl Lyz2-Cre macrophages, the authors demonstrate that USP39 plays an important role in limiting both RNA and DNA viral infection. They further provide evidence that this is through USP39's role in promoting type-I interferon anti-viral responses. Consistently, mouse infection models (VSV and HSV-1) reveal a similar role for USP39 in macrophages in promoting type-I IFN responses to limit viral replication in vivo through analysis of spleen, liver and lung of the mice. Mechanistically, the study provides evidence that USP39 positively regulates RIG-I levels by facilitating splicing of Rig-i pre- mRNA into the mature mRNA. The study claims that USP39 positively regulates the cGAS-STING pathway by stabilising STING through deubiquitylation of Lysine288 and thus preventing proteasomal degradation.

Overall, the study provides novel insights into the role of USP39 in protective anti-viral responses to both RNA and DNA viruses. The data in the manuscript nicely supports this conclusion. The data underpinning the mechanistic aspects of the study is somewhat less convincing and it is not clear why the study focusses specifically on RIG-I and STING as the targets of USP39 - might USP39 not regulate other components of the pathways? The proposed role for USP39 in regulating ubiquitylation of STING needs to be further investigated as the used methodologies to study this are suboptimal and could lead to wrongful conclusions. Specific comments are listed below.

Main Points:

1. Are the authors sure that USP39-regulation of RIG-I levels is the only mechanism by which USP39 regulates the pathway? In Fig 4B MAVS levels also appear reduced in USP39 KO macrophages. Does USP39 also affect splicing of the pre-mRNA encoding MAVS or does it regulate MAVS stability as proposed for STING?

2. Fig 5 I-J. The authors should determine how the re-expression of USP39 impact on splicing of Rig-i mRNA in the macrophages to substantiate their proposed mechanism. Comparison with the USP39-CA in these cells will strengthen their conclusions. The effect of USP39 and USP39-CA on Rig-i mRNA in Figure 5K supports their model but also in these cell model, the effect on Rig-i splicing should be evaluated.

3. The authors claim that USP39 counteracts K48-linked ubiquitination of STING but the experiments shown do not prove this since they IP STING and blot for ubiquitin (total or HA-Ub or K48-Ub). Since IP's are done under non-denaturing conditions, it is not possible to know if the ubiquitin smears observed are conjugated to STING or to other proteins associated with STING. Instead, the experiments should be repeated so ubiquitin is enriched by IP or TUBE pulldown, followed by SDS-PAGE and blotting for STING. If STING migrates as a smear in SDS-PAGE it would strongly indicate that STING is covalently modified with ubiquitin. To further prove that the smear is due to STING ubiquitylation the ubiquitin-enriched sample should treated with recombinant deubiquitinases (e.g. USP21 or OTUB1 (K48-selective)) which will result in a collapse of the smeared STING signal to a single band. The analysis of STING ubiquitination in Figure 7 should also be repeated as described above.

4. The authors conclude that USP39 deubiquitinates K288 on STING to stabilise the protein but the reasoning for this interpretation is not clear. As pointed out by the authors, the ubiquitination signal in Fig 7E is stronger in the samples with K288R STING than in WT STING or the other K>R mutants. However, if K288 is a major ubiquitylation site on STING that is regulated by USP39, the ubiquitination of this STING mutant should be reduced and not increased. Alternatively, if STING is modified on multiple sites, the apparent "length" of the smear should be shorter as the total number of Ub molecules conjugated to each STING K288R molecule would be less than for WT STING. Also, to evaluate the effect of USP39 on the ubiquitination of different STING mutants, the ubiquitination of these should be shown with and without co-expression of USP39. This would enable an assessment of the effect of USP39 expression rather than an assessment of the K>R mutation itself.

Specific Points:

1. Authors should define which murine macrophages are used in their study. Are these BMDMs?

2. In Fig 5 and accompanying text, the authors use the word "Extrons" to describe what I presume are spliced exon-exon boundaries in the mature mRNA or do they refer to refer to something else. Please clarify.

3. Figure 1H/1K. The effect of USP39 KO on IFNbeta production in response to VSV and poly(I:C) is relatively mild (10%-15% reduction), which could indicate that regulation of IFNbeta is not the main function of USP39.

4. Line 249. It is unclear what "compounds" refers to.

5. Fig 6F: The quantification of the protein levels of Flag-STING is unconvincing and appears to contradict the western blots shown: Comparison of the Flag signal in lane 4 vs 1 shows a decrease in signal that is clearly more pronounced than in lane 8 vs 5. However, the graph shoes that USP39 stabilises STING. The quantification should be repeated multiple times and error bars should be included with statistical analysis. Also, the blots are overexposed, which is problematic when trying to quantify the signal.

6. Fig 6G: MG132 treatment of Usp39 fl/fl cells should be included. CHX time course treatments should also be included to determine if, indeed, USP39 increases the half-life of endogenous STING.

---

## [Decision Letter · Decision Letter 2]

12 Jan 2026

Dear Dr chen,

Thank you for your patience while we considered your revised manuscript "USP39 augments antiviral immunity through dual mechanisms of promoting RIG-I pre-mRNA splicing and STING deubiquitination" for publication as a Research Article at PLOS Biology. Your revised study has been evaluated by the PLOS Biology editors, the Academic Editor and the original reviewers.

In light of the reviews, which you will find at the end of this email, we would like to invite you to revise the work to thoroughly address the reviewers' reports.

As you will see below, while we and the reviewers appreciate the thorough revisions done, some concerns remain regarding the demonstration of direct regulation by USP39. Specifically, Reviewer #1 is still not convinced on the mechanism saying that RIG-I is regulated by USP39. Similarly, Reviewer 2 does not think you have properly addressed the previous concern of direct regulation of STING ubiquitination by USP39.

IMPORTANT: after discussion with the Academic Editor, we do require that you perform address experimentally these concerns and provide definitive proof that USP39 regulates RIG-I and STING ubiquitination, as these are the central claims of the study.

Given the extent of revision needed, we cannot make a decision about publication until we have seen the revised manuscript and your response to the reviewers' comments. Your revised manuscript is likely to be sent for further evaluation by all or a subset of the reviewers.

**IMPORTANT - SUBMITTING YOUR REVISION**

*Re-submission Checklist*

*Published Peer Review*

*PLOS Data Policy*

*Blot and Gel Data Policy*

Sincerely,

Melissa

Melissa Vazquez Hernandez, Ph.D.

Associate Editor

PLOS Biology

REVIEWERS' COMMENTS:

Reviewer #1:

The authors added a lot of new data to their manuscript and have addressed most of my comments adequately. However, I remain confused by some of the data, particularly the new data in Fig 4I. Here, the authors over-express using plasmid transfection RIG-I, STING and TBK1. The western blot shows that co-expression of HA-USP39 has no impact on TBK1 protein levels. This is a good control. In contrast, both RIG-I and STING protein levels are increased in HA-USP39 expressing cells. This supports the authors' model for STING (post-translation regulation) but not for RIG-I (regulation of splicing). Presumably, the RIG-I plasmid used here does not contain introns; therefore - if the proposed mechanism is correct - no effect at protein level would be expected in this setting. As such, I am regrettably left unconvinced regarding the mechanism(s) by which USP39 regulates RIG-I.

Reviewer #2:

The authors have addressed most of my concerns and have included in their revised manuscript added important additional experimental data to support their conclusions. However, they have not adequately addressed the concerns raised regarding the direct regulation of K48-liked ubiquitination of STING by USP39. The authors claim that their use of a denaturation IP protocol to evaluate STING ubiquitination proves that the ubiquitin signals are conjugated to STING, referring to one of their previous publications involving NEMO ubiquitination. While the denaturation protocol has been used by some groups to show ubiquitination of NEMO, this remains a contested topic because this is not reproduced when performing pulldown experiments of ubiquitin and blotting for NEMO. Likewise, purification of receptor signalling complexes did not reveal any extensive polyubiquitnation of NEMO. A likely explanation for the discrepancy is that the fact that ubiquitin is not denatured by 1% SDS and, secondly, that ubiqutin-binding domains renature to some extend when the SDS concentration is lowered so that Ub chains can be co-purified.

This may or may not be the case for STING but can the authors be sure that the Ub chains co-purified with STING indeed are conjugated to STING? As suggested previously, the authors could address this by treating their ubiquitin- or STING-enriched samples with recombinant deubiquitinases (e.g. USP21 or OTUB1 (K48-selective)) which will result in a collapse of the smeared STING signal to a single band (USP21) or lower the apparent MW of the "smears" (OTUB1) if the ubiquitin chains indeed are conjugated to STING. See ref: https://www.nature.com/articles/nprot.2015.018 and related articles citing the paper. Along this line, it is curious that the ubiquitin smears in Fig. 7E all have the same migration pattern despite the mutation of specific ubiquitination sites on STING. Would this not suggest that the (majority of the) ubiquitin signals detected comes from ubiquitin chains not directly conjugated to STING? Irrespective, I strongly suggest to either perform the suggested experiments or alternatively, to include a brief discussion of the limitation of this part of the study.

Minor comment: "Extron" instead of "Exon" is still used in Figure S1A,B.

---

## [Editor Report · Decision Letter 3]

14 Apr 2026

Dear Dr chen,

IMPORTANT: please ignore my previous e-mail as that had incorrect information. For your revision, please refer to this e-mail.

Thank you for your patience while we considered your revised manuscript "USP39 augments antiviral immunity through dual mechanisms of promoting RIG-I pre-mRNA splicing and STING deubiquitination" for publication as a Research Article at PLOS Biology. This revised version of your manuscript has been evaluated by the PLOS Biology editors, and the Academic Editor.

Based on our Academic Editor's assessment of your revision, we are likely to accept this manuscript for publication, provided you satisfactorily address the remaining editorial points raised by the reviewers. Please also make sure to address the following data and other policy-related requests.

1) We routinely suggest changes to titles to ensure maximum accessibility for a broad, non-specialist readership, and to ensure they reflect the contents of the paper. In this case, we would suggest a minor edit to the title, as follows. Please ensure you change both the manuscript file and the online submission system, as they need to match for final acceptance:

"USP39 promotes antiviral defense through post-transcriptional control of RIG-I and stabilization of STING"

2) The Ethics statement needs to be a separate, independent (and the first) subheading in the Material & Methods section. It must include the full name of the IACUC/ethics committee that reviewed and approved the animal care and use, as well as the protocol/permit/project license number. https://journals.plos.org/plosbiology/s/ethical-publishing-practice

Please supply the numerical values either in the a supplementary file or as a permanent DOI’d deposition for the following figures:

Figure 1A-N, 2A-J, 3A-HJ-Q, 4G, 5A-N, 6AF, 7AB, S2B-GI, S3C-FH

4) Please cite the location of the data clearly in all relevant main and supplementary Figure legends, e.g. “The data underlying this Figure can be found in S1 Data” or “The data underlying this Figure can be found in https://doi.org/10.5281/zenodo.XXXXX”

5) Please ensure that you are using best practice for statistical reporting and data presentation. These are our guidelines https://journals.plos.org/plosbiology/s/best-practices-in-research-reporting#loc-statistical-reporting and a useful resource on data presentation https://journals.plos.org/plosbiology/article?id=10.1371/journal.pbio.1002128

If you are reporting experiments where n ≤ 5, please plot each individual data point.

6) Thank you for providing the original, uncropped and minimally adjusted images supporting all blot and gel results reported. However, some of the gels still look cropped, as en example see 4B(MAVS) C(HA-USP39) H(STING). If they are uncropped, then it is ok but please clarify. Please carefully read our guidelines for how to prepare and upload this data: https://journals.plos.org/plosbiology/s/figures#loc-blot-and-gel-reporting-requirements

7) Supplementary files (e.g., excel). Please ensure that all data files are uploaded as 'Supporting Information' and are invariably referred to (in the manuscript, figure legends, and the Description field when uploading your files) using the following format verbatim: S1 Data, S2 Data, etc. Multiple panels of a single or even several figures can be included as multiple sheets in one excel file that is saved using exactly the following convention: S1_Data.xlsx (using an underscore).

8) Please add a scale bar in all microscopy pictures

9 Please ensure that your Data Statement in the submission system accurately describes where your data can be found and is in final format, as it will be published as written there

10) Per journal policy, if you have generated any custom code during the course of this investigation, please make it available without restrictions. Please ensure that the code is sufficiently well documented and reusable, and that your Data Statement in the Editorial Manager submission system accurately describes where your code can be found. More information on our Code Policy, what and how to share can be found here: https://journals.plos.org/plosbiology/s/code-availability

We expect to receive your revised manuscript within two weeks.

*Published Peer Review History*

*Press*

Sincerely,

Melissa

Melissa Vazquez Hernandez, Ph.D.

Associate Editor

PLOS Biology

---

## [Editor Report · Decision Letter 4]

27 Apr 2026

Dear Dr chen,

Thank you for the submission of your revised Research Article "USP39 promotes antiviral defense through post-transcriptional control of RIG-I and stabilization of STING" for publication in PLOS Biology. On behalf of my colleagues and the Academic Editor, Sumana Sanyal, I am pleased to say that we can in principle accept your manuscript for publication, provided you address any remaining formatting and reporting issues. These will be detailed in an email you should receive within 2-3 business days from our colleagues in the journal operations team; no action is required from you until then. Please note that we will not be able to formally accept your manuscript and schedule it for publication until you have completed any requested changes.

PRESS

Sincerely,

Melissa

Melissa Vazquez Hernandez, Ph.D., Ph.D.

Associate Editor

PLOS Biology
